



# Biological pumps of carbon, nitrogen, and phosphorus in the northern South China Sea

Jia-Jang Hung[1], Ching-Han Tung[1], Zong-Ying Lin[1], Yuh-Lin Lee Chen[1], Shao-Hung Peng[1], Yen-Huei Lin[1], Li-Shan Tsai

[1]Department of Oceanography, National Sun Yat-sen University, Kaohsiung, Taiwan

*Correspondence to*: Jia-Jang Hung (hungjj@mail.nsysu.edu.tw)

**Abstract.** This paper presents the measured biological pumps (BPs) of carbon (C), nitrogen (N), and phosphorus (P) and their response to seasonal and event-driven oceanographic changes in the northern South China Sea (NSCS). The BP is defined as the sum of active and passive fluxes of biogenic carbon in the surface layer, which may be considered as the central part of marine

carbon cycle. These active and passive fluxes of N and P were also considered to understand stoichiometric flux patterns and the roles of nutrients involved in the BP. The magnitudes of total C, N, and P fluxes were respectively estimated to be 71.9−347 (mean: 163) mg C m$^{-2}$ d$^{-1}$, 13.0−30.5 (mean: 21.6) mg N m$^{-2}$ d$^{-1}$, and 1.02−2.97 (mean: 1.94) mg P m$^{-2}$ d$^{-1}$, which were higher than most previously reported BPs in open oceans, likely because a quarter of the BPs was contributed from active fluxes that were unaccounted for in BPs previously. Moreover, the passive fluxes dominated the BPs and were estimated as 65.3−255 (mean:

125) mg C m$^{-2}$ d$^{-1}$ (76.7% of total C flux), 11.9−23.2 (mean: 17.6) mg N m$^{-2}$ d$^{-1}$ (83.0% of total N flux), and 0.89−1.98 (mean: 1.44) mg P m$^{-2}$ d$^{-1}$ (74.2% of total P flux). Vertical fluxes of dissolved organic C, N, and P generally contributed to less than 5% of passive fluxes. The contrasting patterns of active and passive fluxes found between summer and winter could mainly be attributed to surface warming and stratification in summer and cooling and wind-induced turbulence for pumping nutrients into the euphotic zone in winter. In addition to seasonal variations, the impacts of anticyclonic eddies and internal-wave events on

BP enhancement was apparent in the NSCS. Both active and passive fluxes were likely driven by nutrient availability within the euphotic zone, which was ultimately controlled by the changes in internal and external forcings. The nutrient availability also determined the inventory of chlorophyll *a* and new production, thereby allowing the prediction of active and passive fluxes for unmeasured events. To a first approximation, the SCS may effectively transfer 0.208 Gt C yr$^{-1}$ into the ocean's interior, accounting for approximately 1.89% of the global C flux. The internal forcing and climatic conditions are likely critical factors

in determining the seasonal and event-driven variability of BP in the NSCS.

## 1 Introduction

   It was widely recognized that the global ocean may have absorbed anthropogenic $CO_2$ as large as 50% of total release to the

atmosphere since the industrial revolution began in the middle of 18[th] century (Sabine et al. 2004). The uptake of atmospheric

$CO_2$ by oceans was carried out mainly through the physical pump and biological pump, and both two processes played key
roles in removing carbon from the surface to deep layers of oceans (Ducklow et al., 2001; Boyd et al., 2019). The physical

pump was regarded as the dissolution of atmospheric $CO_2$ into the ocean and then transported into deep oceans through global

circulation (Feely et al. 2001; Toggweiler et al. 2003). Whereas the biological pump (BP) was consisted of active and passive

fluxes of organic carbon synthesized in the euphotic zone and transported out of the surface through zooplankton migration

mediation and gravitational particle settling, respectively, after escaping from respiration and grazing processes in the ocean

surface (Falkowski 1998; Ducklow et al. 2001; Sarmiento and Gruber 2006; Passow and Carlson 2012; Steinberg et al. 2000;

Steinberg and Landry 2017; Archibald et al. 2019). The vertical diffusion flux of dissolved organic carbon (DOC) produced in

the surface has also been thought as a part of passive flux (Ducklow et al. 2001; Steinberg and Landry 2017). The BP was

commonly regarded as an efficient process in downward transfer and storage of carbon dioxide and the critical one in

determining the oceanic carbon cycle and budget (Ducklow et al. 2001; Sarmiento and Gruber 2006; DeVries et al. 2012; Sander

et al. 2014). Thus, Turner (2015) pointed out the BP as one of the most important carbon-involved processes in the world. .

Without BP exporting ~5 Gt C $yr^{-1}$ to the mesopelagic zone, the atmospheric $CO_2$ level would be much higher than they are

today (Parekh et al. 2006; Cavan et al. 2019). Additionally, there was a wide consensus that the marginal sea plays an important

role in modulating the global carbon cycling and fates (Walsh 1991; Liu et al. 2002, 2010; Thomas et al. 2004; Chen and Borges

2009; Dai et al. 2013). Thus, the investigation of biological pump in the large marginal sea appears to be important in increasing

our understanding the global context of oceanic carbon cycling and budgets.

Although the passive transport has long been assumed as the most important process in the transport of carbon from the

surface to deep oceans, the active transport has been considered as an important part of BP showing a substantial proportion

(10–30%) of sinking flux in a variety of oceanographic regimes after 1990s (Longhurst et al. 1989; Dam et al. 1995; Steinberg

et al. 2000; Bianchi et al. 2013). This active transport may not only be important in sustaining the metabolic requirement of

mesopelagic community, but also provide partial energy demand of mesopelagic ecosystem (Robinson et al. 2010; Steinberg et

al. 2008; Burd et al. 2010). Previous studies also showed an imbalance between the heterotrophic activity in mesopelagic waters

and the estimates of carbon supplied by sinking particulate organic carbon (POC), suggesting the importance of diel vertical

migration (DVM) of zooplankton and micronekton in supplying additional demands for microbial growth and respiration in the
mesopelagic zone (Reinthaler et al. 2006; Boyd and Trull 2007; Steinberg et al. 2008; Baltar et al. 2009; Boyd et al., 2019).

Ducklow et al. (2001) as well as Passow and Carlson (2012) have drawn a whole picture of BP illustrating and deciphering the concept and processes of active, passive and DOC fluxes in drawing down atmospheric $CO_2$ and moving various carbon forms from the euphotic zone into the aphotic zone. Although less well documented, the contribution of DOC vertical flux to BP may not be totally neglected particularly in oligotrophic or desert oceans. Previous studies have shown that DOC fluxes may contribute approximately 20–50% of total $C_{org}$ fluxes derived from new production in marginal seas and open oceans (Copin-

Montégut and Avril 1993; Hansell and Carlson 1998, 2001; Avril 2002; Hung et al. 2007; Steinberg and Landry 2017).

Regarding the determination of passive carbon fluxes, sediment traps have been widely used so far to measure the vertical fluxes of POC in various regimes of the ocean (Honjio et al. 2008; Guidi et al. 2015), although they were subject to debates on precision issues (Gardner 2000; Buesseler et al. 2007; Burd et al. 2010). Different approaches may include carbon and nutient budget derivation in the euphotic zone, hydrodynamic-ecosystem model and $^{234}$Th-POC simulation and modelling (Berelson

2001; Ducklow et al. 2001; Buesseler et al. 2009) but they also have certain limitations and won't be discussed here. In terms of active transport, using net captures during day time and night time for sampling DVM zooplankton and micronekton remained the most popular method in estimating active fluxes of carbon and related constitutes (Longhurst et al. 1989; Dam et al. 1995; Steinberg et al. 2000, 2008; Hannides et al. 2009; Takahashi et al. 2009; Yebra et al. 2018). DVM represented the daily ascent of zooplankton and micronekton into the upper layer at dusk and decent into the mesopelagic zone approximately within 600 m

at dawn (Dam et al. 1995; Bianchi et al. 2013). For the reliable estimates of DOC vertical fluxes following the surface accumulation and physical transport may not be a simple work. Hansell and Carlson (2001) and Baetge et al. (2020) have employed the seasonal difference of DOC inventory within surface layers to derive the DOC fluxes through the specified depth (e.g. 100 m). Copin-Montégut and Avril (1993) may be the first persons employing the Fickian-like diffusion law to estimate the DOC vertical flux across a stratified prevailing system.

The South China Sea (SCS) is the largest marginal sea in the world and covers a variety of oceanographic domains including large estuaries, shelf, slope, and a deep central basin, ranging from <100 m to around 5000 m in depth (Shaw and Chao 1994). The northern SCS (NSCS) experiences a strong monsoon influence, the surface circulation is generally clockwise during winter due to prevailing of northeasterly monsoon and anti-clockwise during summer resulting from prevailing southwesterly monsoon

(Wyrtki 1961, Shaw and Chao 1994; Hu et al. 2000). As a result, the physical and biogeochemical conditions of NSCS were

profoundly influenced by seasonal changes of climatic forcing and terrestrial inputs (Shaw and Chao 1994; Dai et al. 2013). The

NSCS is also a hot spot of internal waves generated in the Luzon Strait and transport westward from the Luzon Strait to the

Dongsha-Atoll (DA) continental shelf, causing significant impacts on the DA-associated environments following internal-waves

dissipation and shoaling events (Wang et al. 2007, Li and Farmer 2011; Alford et al. 2015). Therefore, the vertical transfers of

C, N, and P may vary temporally and spatially  under the impacts of atmospheric and oceanic forcings in the NSCS. Despite

many reports have shown a balance or a tiny physical pump of carbon dioxide in most oligotrophic regimes (Zhai et al. 2005,

2012; Dai et al. 2013), the study of BPs is essential and urgent because the limited data have been published so far in realizing

the states and involved processes of BPs in the NSCS. Our ultimate goals focus primarily on understanding the current strengths

of BPs and their controlling mechanisms in the oligotrophic NSCS.

## 2 Materials and methods

### 2.1 Study area and sampling locations

Figure 1 depicts the study area and sampling stations which are located on various regimes in the NSCS. Except for stations

located on the Dongsha-Atoll (DA) associated shelf and upper slope under the influence of internal-wave events, most sampling

stations were located on lower slope and basin regions. To avoid confusion for different names on the same location in different

expeditions, the sampling stations were re-named numerically (Sts. 1−11) to clearly identify them among locations and

expeditions (Table 1). The Station #11 is the Southeast Asian Time-series Study (SEATS) station in the NSCS.








**Figure 1** Maps of the study area and sampling stations. The sampling stations were located mainly in deep-water regions, except for the shallow stations (Stations 7 and 10) close to the Dongsha Atoll. All stations were re-named numerically to avoid confusion with the names originally used in different cruises. For seasonal and spatial comparison, the sampling stations were grouped into two domains, one located in the upper NSCS and one located in the central basin represented by the SEATS (11) station. SEATS denotes the Southeast Asian Time-series Study station in NSCS.





**Table 1** Sampling locations and time periods during various cruises in the northern South China Sea.
Sampling stations were re-named numerically and sampling periods were also noted with
theassociated seasons/events.

| Cruise | Station (Renamed) | Longitude (E) | Latitude (N) | Sampling date | Season (Event) |
|---|---|---|---|---|---|
| ORI-1039 | A (1) | 119°22.67′ | 21°04.08′ | 06/08/2013 | Summer |
| | B (2) | 118°23.86′ | 21°05.26′ | 06/10/2013 | |
| ORI-1059 | 8A (3) | 118°45.08′ | 21°00.56′ | 12/04–11/ | Winter-In[#1] |
| | 7A (3)[$] | 118°10.04′ | 20°00.59.9′ | 2013 | Winter-In[#1] |
| | B4 (4) | 118°00′ | 20°00′ | | Winter-Out[#2] |
| ORI-1074 | A (5) | 117°02.33′ | 20°07.22′ | 05/19/2014 | Later spring |
| | B (6) | 118°29.71′ | 21°04.96′ | 05/20/2014 | |
| ORIII-1773 | S5 (7) | 116°57.15′ | 20°43.84′ | 06/19/2014 | Summer-Internal waves |
| ORI-1082 | B (8) | 118°00.92′ | 21°18.47′ | 07/12/2014 | Summer |
| | C (9) | 117°15.88′ | 21°00.39′ | 07/13/2014 | Summer |
| | D (10) | 116°57.58′ | 20°45.00′ | 07/15/2014 | Summer-Internal waves |
| ORI-708 | | | | 02/16/2004 | Winter[*] |
| ORI-726 | | | | 08/06/2004 | Summer[*] |
| ORI-1184 | SEATS (11) | 115°59.99′ | 17°59.97′ | 11/12/2017 | Fall[@] |
| ORI-1214 | | | | 11/16/2018 | Fall[&] |
| ORI-1240 | | | | 09/22/2019 | Fall[&] |

[#1]In (Inside eddy); [#2]Out (Outside eddy); [#1,2]Vertical POC fluxes were derived from integrated new

productions due to failure of trap recovery. [*]Vertical POC fluxes were derived from integrated new

productions without trap deployment. *Active POC fluxes were derived from DIN and Chlorophyll-*a*

inventories in the euphotic zone. Station 7A[$] (close to 8A) only had an integrated new-production value to




derive vertical POC flux and its POC flux was averaged to the flux on Station 8A to represent the vertical

POC flux within the eddy (Station 3). [@]Passive flux data only; active flux was derived from DIN and

Chlorophyll-a inventories. [&]Active flux data only; passive fluxes were derived from Chlorophyll-*a*

inventories.


## 2.2  Sampling procedures and analytical methods  in seawater

Seawater samplings and electronic data retrieval were carried out on board R/V *Ocean Researcher I* (ORI-1039, ORI-1059,

ORI-1074, ORI-1082), and R/V *Ocean Researcher III* (ORIII-1073, ORIII-1184 and ORIII-1214) (Table 1). Seawater samples

were collected using cleaned Niskin bottle (20 L) mounted on a CTD/Rosette from six  light penetration depths (100%, 46%,

38%, 13%, 5% and 0.6%) in the euphotic zone, and from various depths in the aphotic zone in each station to determine

hydrological and biogeochemical parameters. Seawater temperature (T), salinity (S), depth, and fluorescence were recorded with

CTD and attached probes. Surface and subsurface irradiances were measured with a PAR sensor (OSP2001, Biospherical

Instrument, San Diego, USA). The scientific echo sounder (Simrad EK60) including 38 kHz and 120 kHz was used for recording

the signals of diel migrators located at different depths throughout expeditions. The euphotic zone was recorded as the depth at

which light intensity was 0.6% of surface irradiation (Chen, 2005). The mixed layer depth was estimated from  a difference of

potential density (<0.125) between that of the ocean surface and the bottom of the mixed layer (Monterey and Levitus 1997).

The stratification index (SI) was defined as  the averaged density difference (kg m$^{-4}$) between the surface and a depth of 150 m

(Chen et al., 2014).

The concentration of dissolved oxygen (DO) in retrieved seawater was determined immediately  by following  a method of

direct spectrophotometry of total iodine (Pai et al. 1993). The content of chlorophyll *a* (Chl-*a*) was determined with a fluorometer

(Turner Designs, model AU-10) according to the method of Welschmeyer (1994) after the filtered particulates were extracted

with 90% acetone. Depending on the concentration of particles, various volumes (500−1500 ml) of duplicated seawater samples

were filtered through pre-combusted (at 450 °C, 4 hr) GF/F filters (diameter: 25 mm) to measure dissolved nutrients and

dissolved organic carbon (DOC) in filtrate and particulate organic carbon (POC) in filtered particulates. Dissolved inorganic

nitrogen ($NO_2^- + NO_3^-$, hereafter DIN) and phosphate ($PO_4^{3-}$, hereafter DIP) and silicate ($H_4SiO_4$, hereafter DSi) were

determined colorimetrically (Grasshoff et al. 1983) with a UV-Vis spectrophotometer (Hitachi U-3310) equipped with a module of flow injection analysis for subsurface and deep water samples. DIN and DIP in oligotrophic surface samples were determined by the chemiluminescent method (Garside 1982; Hung et al. 2007) and modified MAGIC method (Thomson-Bulldis and Karl 1998; Hung et al. 2007), respectively. The averaged concentrations and inventories of Chl-*a*, DIN and DIP in the euphotic zone

were estimated from the mean value and trapezoidal integration of all determinants through the euphotic zone, respectively. DOC was measured using a method of  high-temperature catalytic oxidation via  the Shimadzu TOC-5000A analyzer following the established procedures (Hung et al. 2007, 2008). The quality of DOC data was regularly monitored using reference materials (41-44 µM C) provided by Dr. D. A. Hansell from the University of Miami. DON was determined from the difference between dissolved inorganic nitrogen (DIN = $NO_2^-$ + $NO_3^-$) and total dissolved nitrogen (TDN) that was measured with the

chemiluminescence method using an instrument of Anteck Models 771/720 (Hung et al. 2007, 2008). DOP was determined from the difference between DIP and total dissolved phosphorus (TDP) that was measured with UV-persulfate oxidation and colorimetric method (Ridal and Moore 1990).The precision of TDN and TDP analyses was better than ±7% and ±5%, respectively [Hung et al., 2007, 2008].

POC and particulate organic nitrogen (PON) in filtered particulates were determined with an elemental analyzer (Thermo

Scientific Flash 2000) after removal of  carbonate from particulates with 2 M HCl  (Hung et al. 2007, 2008). The analytical precisions of POC and PON were generally < ±0.3 µM C(N) (± 1σ), as evaluated from eight replica samples collected from the same depth. Each biogeochemical parameter was measured in triplicate ensuring the  data quality of analyses in the laboratory (Hung et al. 2007, 2008).

### 2.3 Estimates of active fluxes of carbon, nitrogen and phosphorus

The active flux was determined by collecting diel migrators with a zooplankton net (NORPAC net, 200 µm mesh, d:45-cm, L: 180 cm) coupled with a flow meter (Hydrobios, German) during three day-time (10:00~13:00) and night-time (22:00 ~ 01:00) plankton tows. The difference of integrated biomass profiles in the upper 200-m layer between night and day was regarded as an estimate of the zooplankton and micronekton migrant biomass. The zooplankton net was towed obliquely under 1.5–2.5 knots through the upper layer of 200 m in each sampling time. After collection, the collecting time and water volume were recorded

and the zooplankton and micronekton samples were cleaned with in-situ seawater followed by Milli-Q water and stored in sealed



plastic bags. The samples were frozen immediately with liquid nitrogen and stored at -20 ℃ until further treatment and analyses in the land-based laboratory. In the laboratory, the migrators were size fractionated according to the previously reported methods (Hannides et al., 2009; Al-Mutairi and Landry, 2001) by passing through 0.2, 0.5, 1.0, 2.0, and 5.0-mm sieves. The each size sample was equally split into two parts for experimental purposes. One part was used for immediate analyses of Chl-$a$ and

phaeopigment contents and the remainder was used for species identification (data not reported here) and numeration. The zooplankton and micronekton abundance (A, inds m$^{-3}$) of each class was estimated from total individuals (inds) divided by the flowed water volume (V). The other part was filtered through pre-weighed Nucleopore PC filter (5 μm, 47 mm) to determine the dry-weight (DW) biomass (mg m$^{-3}$) of various planktonic sizes after drying filtered samples in an oven at 60 °C for 3 days. The total migrant biomass was defined by the sum of various sized migrant biomass derived from the difference of sized zooplankton-

micronekton biomass between night-time and day-time tows. The body contents of organic C, N, and P were determined by measuring a specific amount of homogenized dried biomass with same analytical procedures described in the next section for settling materials.

The total active flux reported here includes gut, excretory, respiratory, and mortality fluxes by zooplankton and micronekton (Hannides et al. 2009; Hernández-León et al. 2019). The gut carbon flux was converted from gut Chl-$a$ flux (carbon/Chl-$a$ = 30,

Vidal 1980), and the gut Chl-$a$ flux was estimated from gut contents (gut contents = Chl-$a$ + 1.5 × [phaeopigment]) and gut clearance rate constants (k, h$^{-1}$) according to the methods of Dagg and Wyman (1983) and Dam and Peterson (1988). The Chl-$a$ and phaeopigment contents in zooplankton and micronekton were determined by following the acidification method of Strickland and Parsons (1972). The excretory fluxes of C, N and P were defined as the fluxes of DOC, (DIN+DON), and (DIP+DOP), where DOC, DIN, DON, DIP and DOP fluxes were estimated from migrant DW biomass using empirical allometric

relationships reported by Al-Mutairi and Landry (2001). The excretory rates of ammonia (E$_{DIN}$, μgN ind$^{-1}$ h$^{-1}$) and phosphate (E$_{DIP}$, μgP ind$^{-1}$ h$^{-1}$) were estimated according to Eq. 1 and Eq. 2

ln E$_{DIN}$ = − 2.8900 + 0.7616ln DW + 0.0511T (T is mean temperature at 300−500 m daytime seawater)      (1)

ln E$_{DIP}$ = − 4.3489 + 0.7983ln DW + 0.0285T (T is mean temperature at 300−500 m daytime seawater)      (2)

The magnitude of organic excretion by l migrators was estimated by assuming organic products represent a constant fraction

of the total amount of waste by-products released by migrators at depths (Hannides et al. 2009). The fraction was 0.24 for organic





C (Steinberg et al., 2000), 0.53 for organic N (Le Borgne and Roder, 1997) and 0.47 for organic P (Pomeoy et al., 1963). Thus, the excretory fluxes of dissolved organic C, N, and P (mmol released $m^{-2}$ $d^{-1}$) can be estimated as following equations (Eqs. 3−5)

$$E_{DON} = 0.53/(1 − 0.53) \, E_{DIN} \tag{3}$$

$$E_{DOP} = 0.47/(1 − 0.47) \, E_{DIP} \tag{4}$$

$$E_{DOC} = 0.24/(1 − 0.24) \, R_{DIC} \tag{5}$$

Where $R_{DIC}$ is respiratory $CO_2$ rate ($\mu g$ $CO_2$ evolved $ind^{-1} h^{-1}$) converted from the oxygen consumption rate ($R_O$) ($\ln R_O = −0.2512 + 0.7886 \ln DW + 0.0490T$ (Al-Mutairi and Landry 2001)) assuming a respiratory quotient ($R_Q$) of 0.80 (Hayward 1980).

The respiratory flux was determined using the following equation (Eq. 6) developed by Takahashi et al. (2009)

$$F_r = L_d \times N_i \times RC_i \tag{6}$$

Where $F_r$ is respiratory flux (mg C $m^{-2} d^{-1}$), $L_d$ is length of day time (12 h), $N_i$ is abundance of migrators (inds $m^{-2} d^{-1}$), and $RC_i$ is carbon respiration rate ($\mu g$ C $ind^{-1} h^{-1}$) which is calculated from the empirical relationship ($RC_i = R_O \times R_Q \times 12/22.4$; Takahashi et al., 2009). The mortality flux was estimated from the reported relationship ($F_m = B_i \times M_{deep}$, where $B_i$ is migrant flux through 200 m (mg C $m^{-2} d^{-1}$), and $M_{deep}$ is the mortality rate of migrators (assuming $M_{deep} = 0.01$) (Takahashi et al., 2009).

## 2.4 Experiments on passive fluxes of organic carbon, nitrogen and phosphorus

As an exclusive part of passive flux, the vertical fluxes of settling POC, PON, and particulate organic phosphorus (POP) were determined by using floating sediment traps for particle collection followed by elemental analyses. The traps were deployed generally for three depths (50m, 100m, 150m) in a planned station for approximately 1−3 days, depending on the oceanic condition and ship time availability, to collect sinking particles from upper layers. The sediment-trap array modified from Knauer et al. (1979) consists of two trap sets made from eight Plexiglass tubes (aspect ratio of 9.53) attached to a polypropylene cross frame, similar to those described by Wei et al. (1994), for the depth of 50 m and 100 m, and a commercial sediment trap (PARFLUX Mark8-13, McLane, USA) for a depth of 150 m. All sample tubes were filled with saline seawater to minimize the loss of collected sinking particles. However, no poisons were added to retard bacterial growth and decomposition. In the particular area of DA associated shelf, the PARFLUX trap was attached to the thermistic-fluoroscence string moored at the planned location. After collection, the particulate matter was removed from the PC filter (Polycarbonate, 90 mm, pore size 0.4 µm), washed with Q-water to remove sea salts. After removing swimmers, the particulate matter was freeze-dried to determine




settling fluxes of sinking particles and POC, PON, and POP. In an earlier experiment, vertical fluxes of POC at a depth of 120 m were measured through summer and winter by a deep-moored time-series trap (TECNICAP P.P.S. 3/3) deployed near the SEATS station following the deployed method described in Hung et al (1999) and Chung and Hung (2000).

POC and PON were analyzed by placing collected particulate matter in a silver cup and a few drops of 2 M HCl was added to remove carbonate. The acidified sample was dried in an oven and then determined with an elemental analyzer (Thermo Scientific Flash 2000). Another fraction of particulate matter without treating acid was used for total carbon (TC) analyses. Particulate inorganic carbon (PIC) was the difference between TC and POC. Organic matter content was estimated from POC content by a factor of 2 (Gordon 1970; Monaco et al.1990). Particulate organic phosphorus (POP) was determined from the difference between total particulate phosphorus (PP) and particulate inorganic phosphorus (PIP). PIP was determined by the extraction of particulate matter with 1 M HCl (wt/vol = 50) for 24 hr and the extracted solution was determined by the DIP method described above (Aspila et al. 1976). The concentration of PP was determined by combusting particulate matter at 550 °C for 6 hr followed by extraction and measurement as the same procedures for PIP (Aspila et al. 1976). Analytical uncertainty was < ±6% (n = 6) evaluated from repeated analyses for a coastal sediment. Vertical fluxes of particulate matter, POC, PON, and POP were determined by dividing the collected mass and elements at a specific depth with the trapping area and time period of deployed trap.

Despite of playing minor role in passive fluxes, the downward fluxes of DOC, DON and DOP through a depth of 100 m were estimated from Fick's Law of diffusion (Eq. 7)

$$F_{(100)} = -K_z dC/dz = - [\varepsilon R_f/N^2(p)(1-R_f)][(\overline{C}_1 - \overline{C}_2)/(\overline{z}_2 - \overline{z}_1)] \quad (7)$$

Where $F_{(100)}$ is the flux of DOC (N, P) through a depth of 100 m, $K_z$ is vertical turbulent coefficient, and $dC/dz$ is the gradient of measured parameter concentrations across the boundary. The concentration gradient ($dC/dz$) of DOC (N, P) was calculated from the difference of mean concentrations ($\overline{C}_1 - \overline{C}_2$) divided by the mean depth interval ($\overline{z}_2 - \overline{z}_1$) between two 100-m layers that were above and below the considered boundary (Hung et al., 2007). The $K_z$ was derived from the dissipation rate ($\varepsilon$), the Richardson number ($R_f$) and the square of the Brunt-Väisälä frequency ($N \equiv ((-g/p)(dp/dz))^{1/2}$) at the pycnocline. Therefore, the


EGU Open Access

$K_z$ varies with the inverse of $N^2(p)$, as $\varepsilon$ and $R_f$ are taken as constant values of $10^{-8}$ m$^2$s$^{-3}$ and 0.2, respectively (Copin-Montégut

and Avril, 1993; Doval et al., 2001).

## 2.5  Measurements of primary productivity and new production

Primary productivity (PP) and nitrate-uptake new production (NP) were measured through deck incubation by adding

NaH$^{13}$CO$_3$ and Na$^{15}$NO$_3$ into seawater respectively, following the methods of Chen et al. (2008a). Briefly, water samples were

collected from the same six depths in the euphotic zone.. The collected seawater was transferred immediately into two sets of

three transparent polycarbonate bottles (2.3 L), one set for primary production measurement and the other for new production

measurement. Each set included two light bottles and one dark bottle. The bottles were covered with layers of neutral density

screen to simulate irradiances at the sampling depths and incubated on deck under natural light in incubators circulated with

flow-through surface seawater, starting at approximately 08:00–09:00 h and lasting for 3 h. After incubation, the concentrations

of particulate organic carbon, particulate nitrogen, and the isotopic ratios of $^{13}$C : $^{12}$C and $^{15}$N : $^{14}$N were measured by an automatic

carbon-nitrogen elemental analyser  (ANCA) 20-20 mass spectrometer (Europa Scientific). Details of calculation for PP and NP

can be referred to Chen et al. (2008a).

## 3 Results

### 3.1 Hydrographic characteristics

The oceanographic conditions in the coast-excluded NSCS domains were likely dominated by monsoon-mediated surface

circulation and Kuroshio intrusion (Chen et al., 2005; Dai et al., 2013; Hung et al., 2007, 2020; Liu et al., 2002; Zhai et al., 2005,

2013). In general, a strong northeast monsoon prevails between November and April and a weak southwest monsoon prevails

between June and September leading to a basin-wide cyclonic circulation being dominant in winter and an anticyclonic

circulation being dominant in summer (Shaw and Chao, 1994; Liu et al., 2002; Wong et al., 2007). Thus, Stations 1 and 2 sampled

in summer (July, 2013) exhibited similar distribution (0–300 m) of high surface temperature (T), low surface salinity (S), and

low surface Chl-$a$ concentration with a subsurface maximum (Fig. 2). The mixed layer was shallow (20–27 m) and the T−S

diagram reveals that their characteristics were similar to the typical pattern in South China Sea Water (SCSW; Fig. 3a). Stations

3 and 4 sampled in winter (December, 2013) exhibited low surface T, high surface S, and deeper mixed layer with surface-

elevated Chl-$a$ concentration (Fig. 2). The seawater properties shifted toward the typical features of Kuroshio Water (KW; Fig.



3a), influenced apparently by the intrusion of KW. Stations 3 and 4 were located inside and outside the anticyclonic eddy (Chen

et al., 2015), respectively, with a pronounced deeper mixed layer (160 m vs. 85 m) and higher Chl-*a* at Station 3 than at Station

4. Stations 5 and 6 sampled in later spring (May, 2014) displayed similar patterns with those (T, S, and Chl-*a*) in summer (Stations

1 and 2; Fig. 2). The T-S features belong to certain extents between summer and winter (Fig. 3a).

Station 7 sampled at the location close to the Dongsha Atoll in summer (June, 2014) was influenced by the internal-wave

(IW) shoaling activity, and exhibited low surface T and  high surface S and Chl-*a*, attributed apparently to the upwelling events

(Fig. 2). The T−S diagram also clearly depict the water sourced from subsurface SCSW (Fig. 3b). Stations 8 and 9 sampled in

summer (July, 2014) exhibited the characteristics of SCSW in summer, and the distribution patterns of T, S, Chl-*a* (Fig. 2), and

T−S features (Fig. 3b) were similar to those in Stations 1 and 2. Station 10 sampled in summer (July, 2014) was located at the

same position as Station 7, and exhibited similar features but with slight differences in T, S, Chl-*a*, and T−S properties  (Fig. 2,

Fig. 3b), due to the different upwelling strength. Station 11 (SEATS) sampled in fall (November, 2017) also exhibited high

surface T, low surface S, and moderate surface Chl-*a* with an obvious subsurface maximum (Fig 2). The T−S features shifted

slightly toward the typical features of KW (Fig. 3b). The distribution patterns of T, S, and Chl-*a* in different seasons are also

presented in Figure 4; significant differences in the three parameters were observed between summer and winter, with a deeper

mixed layer, lower surface T, and higher surface Chl-*a* in winter, and vice-versa distributions in summer. Spring and fall were

apparently in transition states between winter and summer (Fig. 4).



**Figure 2** Vertical profiles of temperature, salinity and fluorescence (Chl-*a*) in the upper layer (300 m) of water column for all sampling stations during various expeditions.





**Figure 3** T−S plots for comparing water-column characteristics among stations 1−6 (a) and stations 7−11 (b). Kuroshio and SCS
indicate the typical T−S features of Kuroshio and South China Sea waters, respectively. The Kuroshio and SCS waters
           are typical waters collected from the West Philippine Sea and central SCS, respectively.





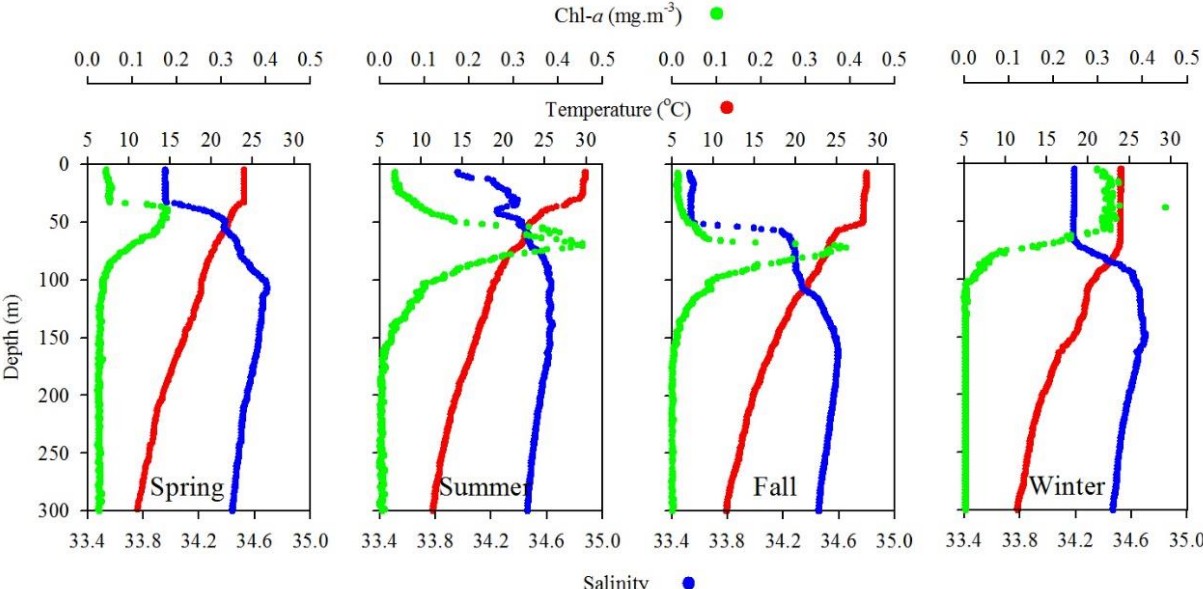

**Figure 4** Comparison of seasonal variations of temperature, salinity, and Chl-*a* profiles in the upper layer (300 m) of water column at SEATS (11) station.


**3.2 Active fluxes of organic carbon, nitrogen and phosphorus**

3.2.1 Evidences of DVM and biomass and abundance of zooplankton and micronekton

The vertical distribution and DVM of the mesopelagic and epipelagic acoustic scattering layers recorded at frequencies of 38 kHz (depth > 1000 m) and 120 kHz (depth approx. ~300 m), respectively, clearly had strong image layers around 400 m

derived at 38 kHz during the day and approximately 100 m derived at 120 kHz during the night (Figure not shown). This finding indicates that the vertical migrators were located largely at a depth of 400 m during the daytime and migrated to approximately 100 m during the night time. Sequential tows performed at eight time intervals (17:00, 21:00, 01:00, 03:00, 06:00, 09:00, 12:00, and 15:00) at the top 200 m revealed the largest mesozooplankton biomass (2021 mg m$^{-2}$) and abundance (354 inds m$^{-3}$) during the night (01:00) but the lowest biomass (1480 mg m$^{-2}$) and abundance (270 inds m$^{-3}$) during the day (12:00; Supplementary

Fig. S1). Higher mesozooplankton biomass and abundance were observed in night tows than in day tows for all size classes; the occurrence of small mesozooplankton (0.2–2.0 mm) was generally higher than that of large mesozooplankton (2.0–5.0 mm), except for the highest occurrence of large (0.2–5.0 mm) mesozooplankton in winter (Table 2). However, the magnitude of migrant biomass (night minus day) was usually the largest for the 2.0–5.0 mm class, except during an internal-wave event in summer (Table 2). The total migrant biomass (sum of all sizes) was 474 mg m$^{-2}$ in late spring, ranged from 235 to 418 (mean:

327) mg m$^{-2}$ in summer, was 635 mg m$^{-2}$ in winter with an anticyclonic event, and ranged from 158 to 189 (mean: 174 ) mg m$^{-2}$





during fall at SEATS station (Table 2). An elevated biomass of 997 mg m$^{-2}$ was observed in the internal-wave influencing fields in summer (Table 2). The night/day ratio of migrant biomass was higher for large mesozooplankton (2.15−3.12 for size 2.0−5.0 mm) than for small mesozooplankton (1.21−2.09 for size 0.2−0.5−1.0 mm), coincident with the size distribution of migrant biomass (Table 2). This implied that larger migrators might play crucial roles than smaller migrators in determining the vertical transport of materials and elements.

**Table 2** A list of mesozooplankton biomass and migrant biomass in various sizes collected from night and day tows, and night/day (N:D) biomass ratio during different seasons and events

| Season/size fraction | Dry biomass (mg m$^{-2}$) | | | |
|---|---|---|---|---|
| | Night | Day | N:D | Migrant biomass |
| Summer [Grand average from OR1-1039 (2013), OR1-1074 (2014), and OR1-1082 (2014)] | | | | |
| 0.2-0.5 mm | 308±97 | 249±93 | 1.24 | 59.3 |
| 0.5-1 mm | 319±164 | 252±142 | 1.27 | 66.9 |
| 1-2 mm | 316±205 | 211±153 | 1.5 | 105 |
| 2-5 mm | 243±118 | 99±60 | 2.47 | 145 |
| total (>0.2 mm) | 1186±304 | 811±236 | 1.46 | 376 |
| Winter [Anticyclonic-eddy event OR1-1059 (2013)] | | | | |
| 0.2-0.5 mm | 271 | 132 | 2.05 | 139 |
| 0.5-1 mm | 196 | 94 | 2.09 | 102 |
| 1-2 mm | 267 | 69 | 3.87 | 198 |
| 2-5 mm | 336 | 140 | 2.39 | 196 |
| total (>0.2 mm) | 1070 | 435 | 2.46 | 635 |
| Summer [Grand average from OR3-1773 (2014) and OR1-1082 (2014) in internal-wave influencing fields] | | | | |
| 0.2-0.5 mm | 1061±387 | 811±388 | 1.31 | 250 |
| 0.5-1 mm | 1008±401 | 775±416 | 1.30 | 233 |
| 1-2 mm | 1018±393 | 742±213 | 1.37 | 276 |





| | | | | |
|---|---|---|---|---|
| 2-5 mm | 466±209 | 229±153 | 2.04 | 237 |
| total (>0.2 mm) | 3554±713 | 2557±667 | 1.39 | 997 |
| Fall [Grand average from OR1-1214 (2018) and OR1-1240 (2019)] | | | | |
| 0.2-0.5mm | 123±57 | 101±50 | 1.22 | 22.1 |
| 0.5-1mm | 168±2 | 132±8 | 1.27 | 36.0 |
| 1-2mm | 91±32 | 44±40 | 2.07 | 47.3 |
| 2-5mm | 119±31 | 51±1 | 2.34 | 68.2 |
| total (>0.2mm) | 501±60 | 327±82 | 1.53 | 174 |


### 3.2.2 Elemental composition of mesozooplankton

The measurement of elemental contents of mesozooplankton is essential for determining active fluxes of carbon (C), nitrogen (N), and phosphorus (P). The planktonic contents of C, N, and P were 37.4±4.34%, 7.86±1.29%, and 0.76±0.43%, respectively, which did not significantly differ between day-time and night-time tows in summer. In general, C and N contents

were higher in smaller mesozooplankton (1.0−2.0 and 0.5−1.0 mm) than in larger mesozooplankton (2.0−5.0 mm), but the P content increased with an increase in mesozooplankton size. The C, N, and P contents were respectively 33.2±10.3%, 6.21±2.10%, and 1.06±0.69% in winter, with an occurrence of anticyclonic eddy; 39.4±3.67%, 7.88±1.02%, and 0.91±0.36% in internal-wave influencing fields in summer; and 40.4±1.13%, 8.92±0.43%, and 0.60±0.08% in fall at the SEATS station. The C and N contents were similar to those reports previously (35.6%−40%, Parsons et al., 1979; Dam and Peterson, 1993; Kobari et

al., 2013; and 9%, Peters and Downing, 1984, respectively). The molar ratios of C:N, C:P, and N:P varied seasonally, ranging from 5.29 to 5.80 (5.55±0.16), 79.7 to 162 (131±30), and 15.1 to 29.6 (23.6±5.05), respectively, in summer. The elemental ratios of C:N, C:P, and N:P were 4.97−7.42 (6.33±0.71), 45.3−211 (102±50.6), and 9.12−35.3 (16.0±8.2), respectively, in winter, and 5.31−6.23 (5.84±0.27), 76.8−134 (111±29.9), and 3.5−22.0 (18.9±3.16), respectively, in summer in the internal-wave influencing fields. Moreover, they were 4.15−5.49 (5.2±0.27), 139−215 (176±31), and 25.2−40.6 (33.2±6.29) in fall at the

SEATS station. The elemental ratios of C:P and N:P exhibited greater variation than C:N, which likely resulted from the large variation in P content. The elemental composition, however, was comparable with that found in the ALOHA station ($C_{88}N_{18}P_1$; Hannides et al., 2009), Baltic Sea ($C_{41}N_7P_1-C_{144}N_{24}P_1$; Pertola et al., 2002), and Norwegian Fjord ($C_{63}N_8P_1-C_{348}N_{38}P_1$; Gismervik, 1997). Our C:N:P ratios were apparently lower than the Redfield ratio ($C_{106}N_{16}P_1$), likely because of the relatively high N and P contents in mesozooplankton compared with phytoplankton.

### 3.2.3 Active fluxes of C, N and P



Active fluxes of C, N, and P were estimated as the sum of respiratory, gut, excretory, and mortality fluxes for mesozooplankton of various size fractions, and the original on component fluxes are presented in Supplementary Table S1. In terms of C flux, the respiratory flux was the most dominant, followed by gut flux, excretory DOC flux, and mortality flux. By contrast, the N and P fluxes were derived mainly from excretory and mortality fluxes, and the excretory fluxes were considerably

higher than the mortality fluxes. In general, the respiratory, gut, and excretory C fluxes decreased with an increase in the size fractions with a few exceptions (Supplementary Table S1). However, the excretory and mortality fluxes of N and P did not exhibit a consistent relationship with size fractions (Supplementary Table S1). Overall, the active C flux was mainly accounted for by the respiration flux (49.4%−75.8%) and the least by the mortality flux (8.99%−13.4%); those results were comparable to those of the proportion of respiration flux contributing to active flux in the western equatorial Pacific (54.6%; Hidaka et al.,

2001), subtropical Pacific Ocean (61.8%−63.0%; Kobari et al., 2013), and Sargasso Sea (BATS Station, 75%; Steinberg et al., 2000).

Resolving spatial and seasonal variations in active fluxes in the NSCS is difficult because of unsuccessful sampling at certain stations and cruises. Nevertheless, for the first-order approximation, the active fluxes that could not be measured were estimated using the empirical relationship established from the experimental data of active fluxes and Chl-*a* inventories (Fig. 5).

Thus, the compiled active fluxes of C, N, and P were 7.69–93.4 mg C m$^{-2}$ d$^{-1}$, 1.06–7.26 mg N m$^{-2}$ d$^{-1}$, 0.13–0.99 mg P m$^{-2}$ d$^{-1}$, respectively (Fig. 6). The flux distribution was the highest in summer due to the impact of internal-wave upwelling, followed by in winter with an anticyclonic eddy, and finally, in summer with a calm oceanic condition. The smallest values were found in the fall season under relatively calm condition (Fig. 3) on the central basin (SEATS, St. 11), which is far from land sources.

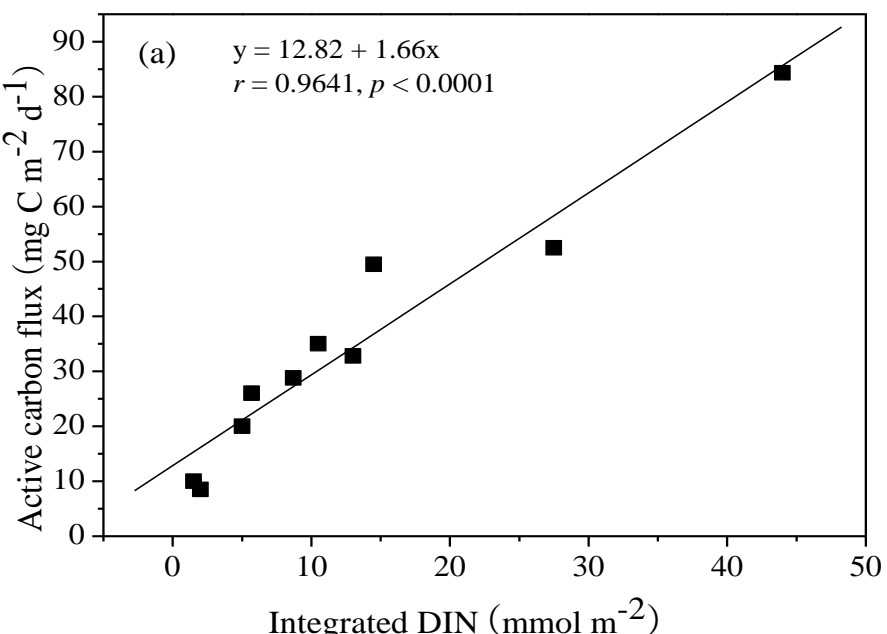

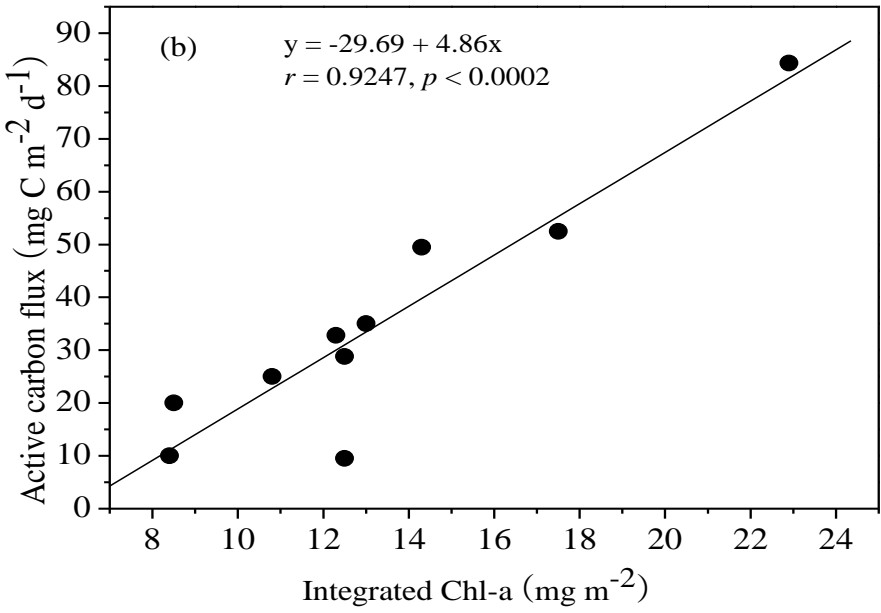


**Figure 5** Empirical relationship between active carbon fluxes and DIN inventories in the euphotic zone (a) and between active carbon fluxes and Chl-*a* inventories in the euphotic zone (b). The statistic correlations were established from collected data in various expeditions.

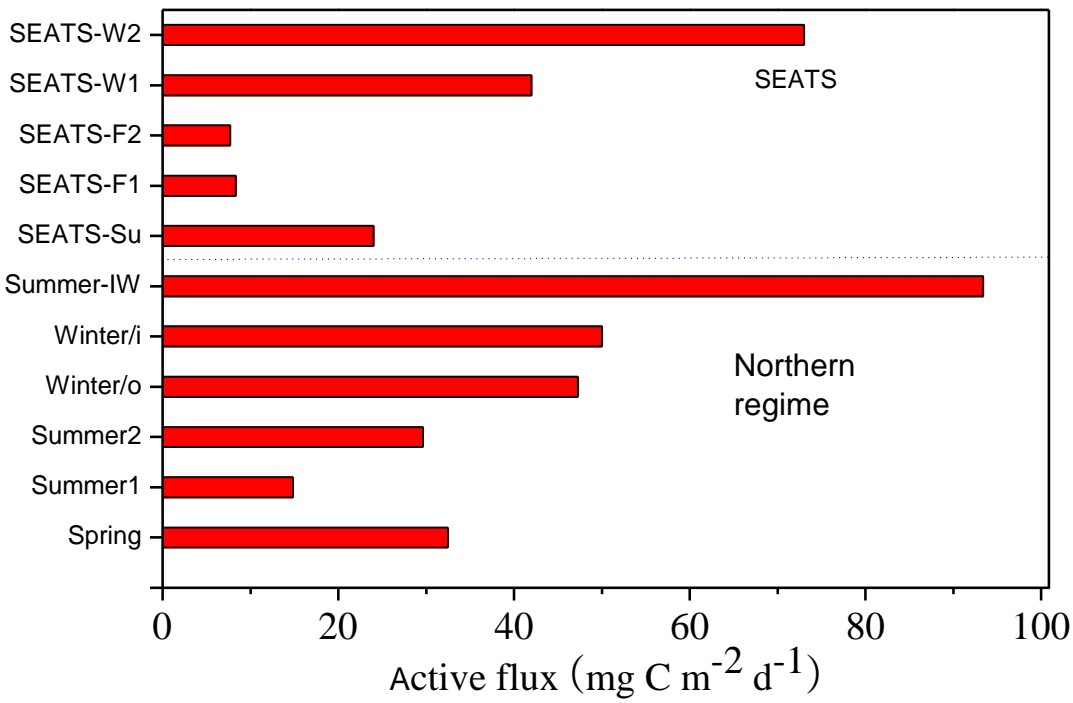




**Figure 6** Comparisons between seasonal and spatial active carbon fluxes in the NSCS. The active fluxes were geographically grouped as the central basin represented by the SEATS station and the northern regime for other sampling locations. The SEATS active fluxes were estimated using the empirical relationship between active fluxes and inventories of Chl-*a* except for SEATS-F and 2 (fall season), which were derived from experimental data. The data of the northern regime were all experimental data, except for the Winter/o (outside the eddy) datum derived from the empirical relationship between active fluxes and Chl-*a* inventories. Winter/i (inside the eddy); Summer-IW (internal waves); SEATS-Su (summer); SEATS-W (winter); SEATS-F (fall).

### 3.3 Passive fluxes of C, N, and P

3.3.1 Vertical fluxes of POC, PON, and POP

Vertical fluxes of POC, PON, and POP appeared to decrease with an increase in depth from 50 to 150 m, likely due to the increased decomposition of organic matter with increasing depth (Table 3). Because most euphotic zones were located at depths between 50 and 100 m, vertical fluxes through a depth 100 m were considered the measures of passive fluxes. To obtain a comprehensive understanding and for comparison, some fluxes through a depth of 100 m were obtained through prediction based on the euphotic-layer inventories of new production, DIN, and Chl-*a* (see the Discussion section) for stations that exhibit trap recovery failure or those with no trap deployment in previous studies. Vertical POC fluxes through a depth of 100 m ranged from $64.3 \pm 1.47$ mg C m$^{-2}$ d$^{-1}$ in regular summer to 165 mg C m$^{-2}$ d$^{-1}$ in regular winter. The flux increased to $156 \pm 15.9$ mg C m$^{-2}$ d$^{-1}$ in summer with the internal-wave upwelling field and to $175 \pm 3.5$ mg C m$^{-2}$ d$^{-1}$ in winter within the anticyclonic eddy (Table 3, Fig. 7). At the SEATS station located in the central basin, the POC fluxes ranged from 51.4 mg C m$^{-2}$ d$^{-1}$ during fall to 100 mg C m$^{-2}$ d$^{-1}$ during winter (Table 3). Additional data obtained from previous sequentially moored traps at the SEATS station at a depth of 120 m revealed extremely high fluxes (199–254 mg C m$^{-2}$ d$^{-1}$) in winter (SEATS-W2, SEATS-W3; Fig. 7). Although data on PON and POP fluxes were limited, the data predicted after the addition of POC:PON and POC:POP ratios the seasonal and event-effected patterns followed apparently with the variability of POC fluxes (Table 3).

The molar ratios of POC:PON ranged from $5.65 \pm 0.20$ (at 50 m) to $8.00 \pm 0.15$ (at 100 m), with an overall value of approximately $6.84 \pm 0.60$ (data not shown). The C:N ratio increased slightly from 50 to 150 m, likely attributed to the rapid decay of PON over POC with increasing depth. The mean ratio was close to the Redfield ratio (6.6; Redfield, 1958), indicating a relatively low contribution of lithogenic POC sources. The molar ratios of POC:POP ranged from $152 \pm 1.57$ (at 50 m) to $243 \pm 15.3$ (at 150 m), with an overall value of approximately $194 \pm 9.5$. The increase in C:P ratios with increasing depth was more pronounced than that of C:N ratios, indicating that POP was more labile PON in settling organic matter. The C:N and C:P ratios were applied to the estimation of the PON and POP fluxes not obtained from the measured POC fluxes presented in Table 3.

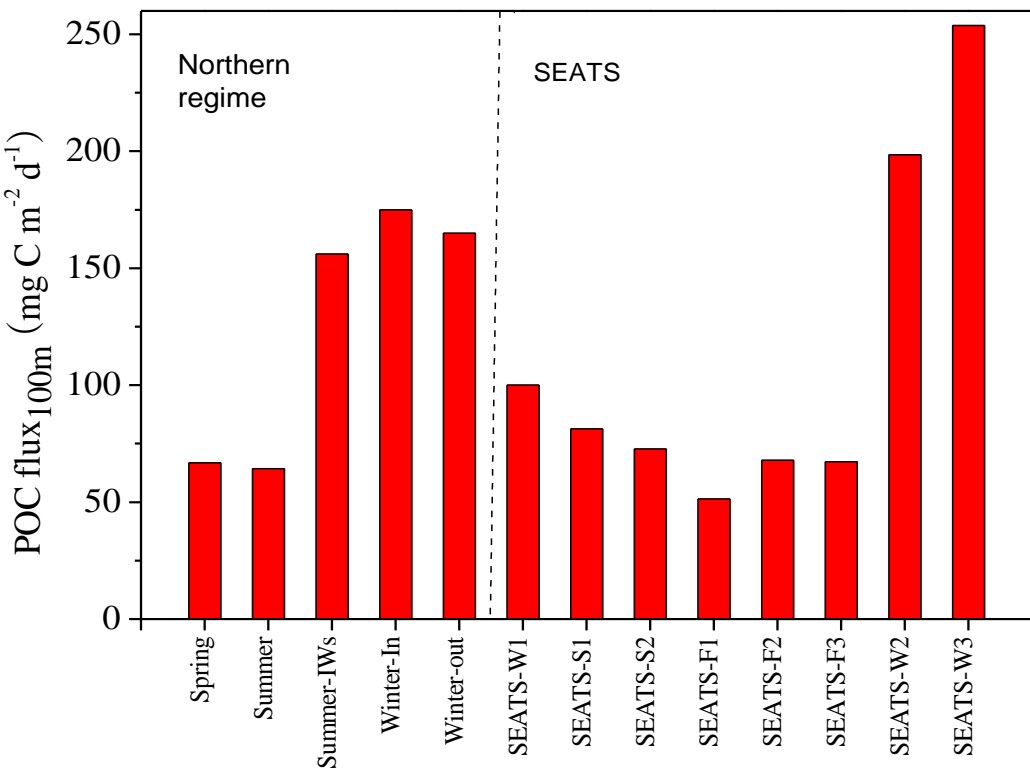

**Figure 7** Seasonal variations in vertical POC fluxes in the SEATS-excluded region (left side) and SEATS station (right side).
Summer-IWs denotes the internal-wave event in summer; Winter-In denotes values inside the anticyclonic eddy in winter;
Winter-out denotes values outside the anticyclonic eddy in winter. SEATS-W1, S1, S2, F1, F2, F3, W2, and W3 represent
various samplings at winter (W), summer (S) and fall (F) seasons at the SEATS station. SEATS-W2 and W3 data were
obtained from the bottom-moored traps at a depth of 120 m (see Fig. 11). Other SEATS data were derived from integrating
data of the new production and Chl-*a* (see Figs. 9 and 10) except for data of SEATS-F1, which were obtained from the
deployed floating traps.





**Table 3** A list of measured and predicted fluxes of total mass, POC, PON, and POP in various sampling seasons and oceanic events in NSCS.

| Seasons/Events | Depth (m) | Mass flux (mg m$^{-2}$ d$^{-1}$) | POC flux (mg C m$^{-2}$ d$^{-1}$) | PON flux (mg N m$^{-2}$ d$^{-1}$) | POP flux (mg P m$^{-2}$ d$^{-1}$) |
|---|---|---|---|---|---|
| Late spring | 50 | 270±22.3 | 101±10.7 | 20.5±2.61 | 1.71±0.16 |
| (ORI-1074, 2014) | 100 | 221±28.8 | 66.8±1.29 | 12.8±0.38 | 0.99±0.07 |
|  | 150 | 99.1±14.1 | 21.6±2.06 | 3.31±0.52 | 0.24±0.04 |
| Summer | 50 | 286±8.20 | 104±13.4 | 21.5±2.01 | 1.71±0.16 |
| (ORI-1039, 2013; ORI-1082, 2014) | 100 | 218±25.0 | 64.3±1.47 | 12.1±0.47 | 0.93±0.04 |
|  | 150 | 89.4±4.01 | 19.6±6.06 | 2.85±0.82 | 0.21±0.06 |
| Internal waves (summer, ORI-1082, ORIII-1773) | 100 | 334±33.0 | 156±15.9 | 21.2±1.68 | 1.79±0.19 |
| Winter (ORI-1059, 2013; inside eddy) | 100 | – | 175±35[#] | (25.9±5.1) | (0.90±0.18) |
| Winter(ORI-1059, 2013; outside eddy) | 100 | – | 165[#] | (24.1) | (0.84) |
| SEATS (winter, ORI-708. 2004) | 100 | – | 100[#] | (14.6) | (0.52) |
| SEATS (summer, ORI-722, 2004) | 100 | – | 81.3[#] | (11.9) | (0.42) |
| SEATS (summer, ORI-726, 2004) | 100 | – | 72.7[#] | (10.6) | (0.37) |
| SEATS (Fall, ORI-1184, 2017) | 50 | 230 | 61.9 | 9.46 | 0.85 |
|  | 100 | 201 | 51.4 | 7.00 | 0.61 |
| SEATS (Fall, ORI-1214, 2018) | 100 | – | 67.9[@] | (9.93) | (0.35) |
| SEATS (Fall, ORI-1240, 2019) | 100 | – | 85.5[@] | (12.5) | (0.44) |





[#]POC fluxes were derived from integrated new production (see Fig. 9); [@]POC fluxes were derived from Chl-*a* inventories
in the euphotic zone (see Fig. 9a); PON and POP fluxes in parentheses were estimated from POC fluxes and C:N and C:P
ratios.


### 3.3.2 Vertical fluxes of DOC and DON

Although the data on DOC and DON fluxes through a depth of 100 m were limited, for first-order approximation,
considering the contribution of DOC and DON fluxes to passive carbon and nitrogen fluxes was essential. In general, the vertical
fluxes of DOC and DON likely increased from a depth of 50 to 150 m, ranging from $0.71\pm0.68$ mg C m$^{-2}$ d$^{-1}$ at 50 m to $1.71\pm0.01$
mg C m$^{-2}$ d$^{-1}$ at 150 m in spring and from $0.78\pm0.52$ mg C m$^{-2}$ d$^{-1}$ at 50 m to $1.29\pm0.15$ mg C m$^{-2}$ d$^{-1}$ at 150 m in summer
(Supplementary Table S2). Vertical fluxes of DOC through a depth of 100 m were $1.13\pm0.03$ mg C m$^{-2}$ d$^{-1}$ in spring and
$1.10\pm0.13$ mg C m$^{-2}$ d$^{-1}$ in summer. The DON fluxes ranged from $0.08\pm0.06$ mg N m$^{-2}$ d$^{-1}$ at 50 m to $0.35\pm0.02$ mg N m$^{-2}$ d$^{-1}$
at 150 m in spring and from $0.06\pm0.06$ mg N m$^{-2}$ d$^{-1}$ at 50 m to $0.10\pm0.08$ mg N m$^{-2}$ d$^{-1}$ at 150 m in summer (Supplementary
Table S2). Vertical fluxes of DON through a depth of 100 m were $0.22\pm0.07$ mg N m$^{-2}$ d$^{-1}$ in spring and $0.09\pm0.06$ mg N m$^{-2}$
d$^{-1}$ in summer. The DOC and DON fluxes through a depth of 100 m increased to $1.57\pm1.07$ mg C m$^{-2}$ d$^{-1}$ and $0.36\pm0.25$ mg N
m$^{-2}$ d$^{-1}$, respectively, during the summer influenced by internal-wave events. However, vertical flux data of DOC and DON in
winter could not be obtained.

## 4   Discussion

### 4.1 Regulation of active C, N, and P fluxes in the NSCS

Both migrant biomass and migratory fluxes of C, N, and P varied with seasons, locations, and oceanic events. Although
determined independently, migrant biomass and active CNP fluxes coincidently varied with seasons and oceanic events. As a
result, migrant biomass was closely correlated with migratory fluxes of C ($r = 0.8343$, $p < 0.0001$), N ($r = 0.7800$, $p < 0.0001$),
and P ($r = 0.8259$, $p < 0.0001$; Fig. 8), indicating the crucial role of migrant biomass in determining the magnitudes of active C,
N, and P fluxes. The increase in migrant biomass apparently increased the predation of phytoplankton during the night in the
upper layers, which likely enhanced the metabolic and clearance rates of migrators during the daytime in mesopelagic zones
because the two rates dominated the magnitudes of active fluxes (Supplementary Table S1). Moreover, the larger migrators,
particularly those of sizes 2−5 mm, appeared to be dominant in transporting C, N, and P into mesopelagic zones (Table 2), which
is consistent with the results of Valencia et al. (2018) who reported  2−5 mm migrators as the major group in determining active
fluxes at station ALOHA, North Pacific Subtropical Gyre. Steinberg and Landry (2017) compiled the data of migrant biomass



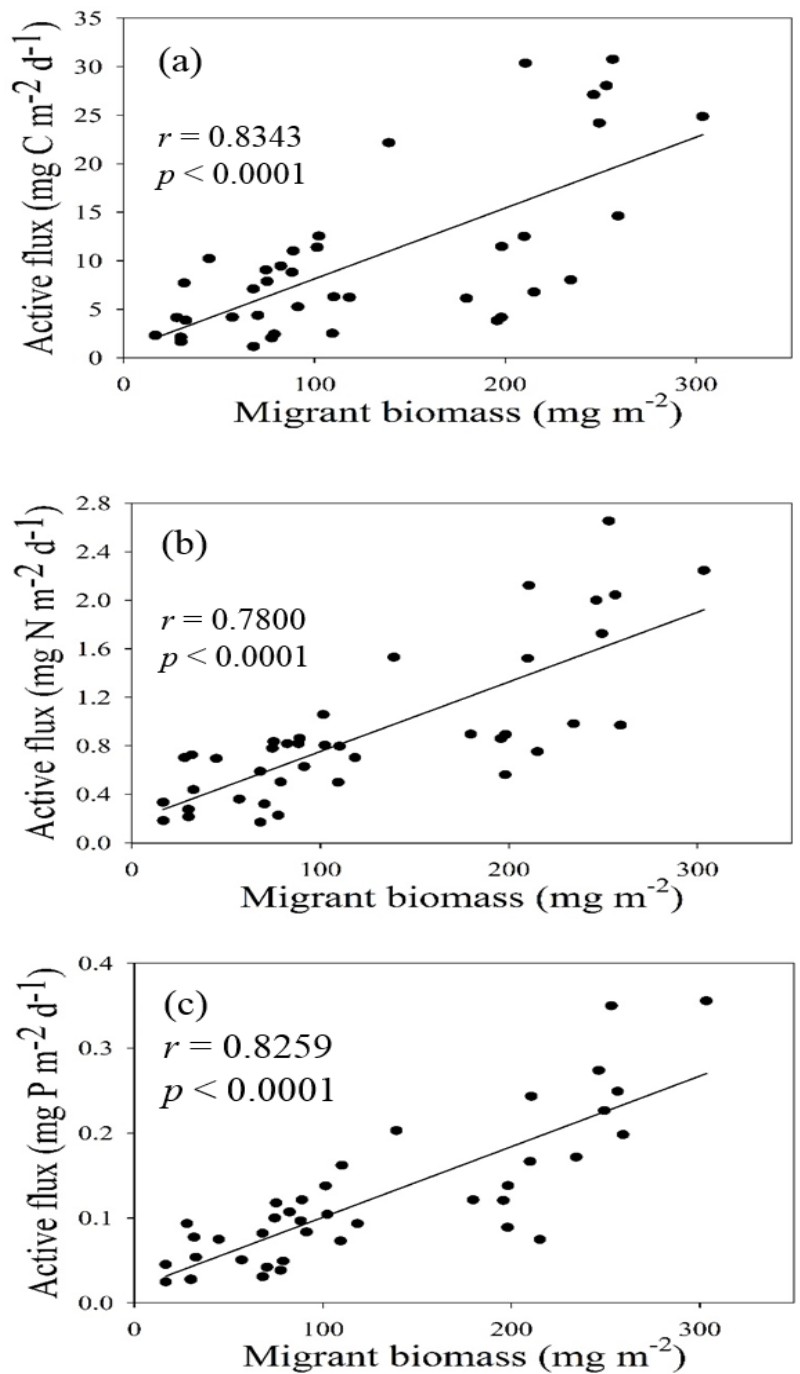

**Figure 8** Plots of statistic correlations between migrant biomass and active carbon
fluxes (a), active nitrogen fluxes (b), and active phosphorus fluxes (c).

and respiratory carbon fluxes collected from various locations in the North Atlantic and Pacific Oceans and demonstrated an

increase in respiratory carbon fluxes with an increase in migrant biomass (positive correlation). In addition, with an increase in

respiratory carbon fluxes, the equivalent fraction of vertical POC fluxes measured by traps from epipelagic zones (100−200 m)

also increased. Although the oceanic conditions may influence the community structure, size distribution, and migrant biomass

leading to changes in active-flux magnitudes (Valencia et al., 2018), our data indicated that the 2−5 mm class exhibited the

highest N:D biomass ratios and migrant biomass in both summer and winter with contrasting oceanic conditions in the NSCS,

implying the dominant role of 2−5 mm migrators in determining migratory fluxes in the subtropical-tropical ocean.

The NSCS experiences contrasting atmospheric and oceanic forcings between the winter and summer including most of the

time during spring and fall (Liu et al., 2002; Hung et al., 2020). In general, the upper-ocean stratification progressed from spring

to summer (SI, 0.025−0.04 kg m$^{-4}$) with an increase in temperature and weak southwesterly monsoon winds, after which the

stratification began to erode from fall to winter (SI, < 0.01 kg m$^{-4}$) due to surface-water cooling and the prevailing northeasterly

monsoon winds. The subsurface nutrient pumping through the eutrophic base may intensify the entry into the winter season.

Thus, the discrete contents and inventories of nutrients and Chl-$a$ in the euphotic zone were considerably higher in winter than

in summer in the NSCS, excluding the coastal and shelf zones reported in our previous studies (Hung et al., 2007; 2020; Chen

et al., 2008, 2014) and in the current experiments. To obtain a complete data set of active fluxes for seasonal comparison, the

flux data that could not be collected were derived from the data of Chl-$a$ and DIN inventories using appropriate correlations

between active carbon fluxes and Chl-$a$ inventories ($r = 0.9247$, $p$ <0.002; Fig. 5a) and between active carbon fluxes and DIN

inventories ($r = 0.9641$, $p$ <0.0001; Fig. 5b) constructed from the successfully collected data in the current study. These

empirical relationships may also indicate that the active fluxes were driven by the availability of nutrients (DIN) in the euphotic

zone, which in turn determined Chl-$a$ inventories because of a significant correlation between integrated DIN and integrated

Chl-$a$ ($r = 0.9479$, $p < 0.0001$).

In the northern regime, active fluxes were generally higher in winter than in spring and summer, likely due to the increase

of nutrient pumping in winter. In addition, the active flux was slightly higher in the region within the anticyclonic eddy (St. 3)

than the in the region located outside the eddy (St. 4; Fig. 5), as a result of the eddy-enhanced nutrient pumping to the euphotic

zone. Chen et al. (2015) demonstrated that this anticyclonic eddy occurring during winter was characterized by a deep mixed

layer of up to 140−180 m and the concentration of nitrate and Chl-*a* increased in the top water column (0−200 m), resulting in an increase in primary productivity and new production. Thus, the nutrient pumping in the euphotic zone appears to be the major driver enhancing the active carbon fluxes in winter and in anticyclonic eddy-driven events. The extremely high active carbon flux that occurred in the internal-wave influencing field near the Dongsha Atoll was also attributed to the strong nutrient upwelling caused by the elevation of waves despite of the summer season conditions (Hung et al., 2021). At the SEATS station located on the central basin, the active carbon fluxes were not necessarily lower than those found in respective seasons in the northern regime, although the lowest fluxes were noted during the fall season (Fig. 6). Similarly, the carbon fluxes were considerably higher in winter than in other seasons at the SEATS station, likely attributable to the abovementioned mechanism.

Data on active nitrogen and phosphorus fluxes in the NSCS are limited. To a first approximation, active nitrogen and phosphorus fluxes were derived from excretory and mortality fluxes; they respectively ranged from 1.06 mg N m$^{-2}$ d$^{-1}$ and 0.13 mg P m$^{-2}$ d$^{-1}$ during fall at SEATS station to 3.21 mg N m$^{-2}$ d$^{-1}$ and 0.40 mg P m$^{-2}$ d$^{-1}$ during spring, 1.77 mg N m$^{-2}$ d$^{-1}$ and 0.33 mg P m$^{-2}$ d$^{-1}$ during summer, 3.51 mg N m$^{-2}$ d$^{-1}$ and 0.57 mg P m$^{-2}$ d$^{-1}$ during the winter-eddy event, and 7.26 mg N m$^{-2}$ d$^{-1}$ and 1.08 mg P m$^{-2}$ d$^{-1}$ during the summer-IWs event. In general, the distribution of active nitrogen and phosphorus fluxes followed the seasonal patterns of active carbon fluxes. The C:N ratios of active fluxes ranged from 6.9 (fall) to 14.2 (winter; mean: 10.6) and the C:P ratio ranged from 55.7 (fall) to 87.7 (winter; mean: 72.9). The C:N and C:P ratios appeared to increase with an increase in active fluxes, likely caused by the increased contribution of respiration and gut fluxes to active fluxes, and the respiration and gut fluxes did not include nitrogen and phosphorus fluxes. Moreover, higher respiration and gut fluxes occurred in winter than in summer. The C:N and C:P ratios of active fluxes were respectively higher and lower than the C:N and C:P ratios of particulate vertical fluxes, the major component of passive fluxes.

**4.2  Controlling mechanisms of passive fluxes of C, N, and P**

Vertical POC fluxes varied with seasons and locations (Fig. 7), likely because of a pronounced difference in hydrographic and biogeochemical conditions between summer and winter. The upper water column has been widely reported to undergo stratification and experience restricted nutrient availability in summer; however, in winter surface stratification was eroded and nutrient availability increased, leading to enhanced primary productivity and new production (Figs. 2&4; Chen, 2005; Chen et al., 2008a; 2014; Dai et al., 2013; Zhai et al., 2013: Hung et al., 2020). By combining the previous and current measurements,




particularly our coauthor's (Chen, Y.-L.) new-production data, we found a striking relationship ($r = 0.8502$, $p < 0.02$) between

integrated new productions and vertical POC fluxes through a depth of 100 m (Fig. 9). Vertical POC fluxes have also been

efficiently predicted from primary production ($R^2 = 0.69–0.97$) in various regimes of the ocean (Baltzer et al., 1984; Pace et al.,

1987). However, Karl et al. (1996) later found an inverse correlation between POC fluxes and primary production during the

ENSO period at ALOHA station. Under the oceanographic paradigm, new production is a significant contributor of primary

productivity and the export production; therefore, a strong correlation between vertical POC fluxes and new productions is

expected. By using this empirical relationship, the data of vertical POC fluxes that could not be collected in this study can be

predicted on the basis of the new production data and the more efficient data set of vertical fluxes can be used for spatial and

seasonal comparisons.

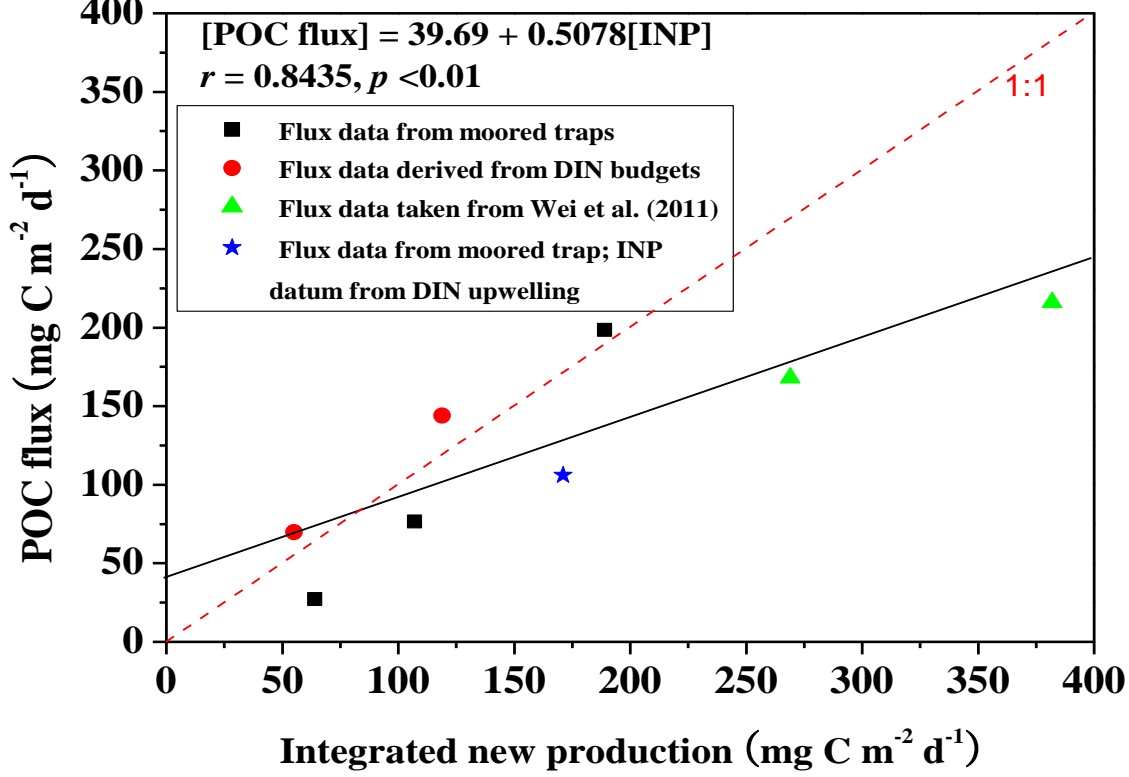

**Figure 9** Scatter plots depicting the relationship between integrated new production (INP) and POC fluxes through a depth

of 100 m at the SEATS station, except for a datum (star symbol) derived from the station near the Dongsha Atoll (Hung

et al., 2021). INP data were adapted from Chen et al. (2007, 2008a, 2014) except for a datum derived from Hung et al.





(2021). Data of POC fluxes through 100−120 m were derived from the moored trap (Tsai, 2007; Hung et al., 2021) and floating traps (Wei et al., 2010), except for two data items derived from DIN budgets (Hung et al., 2007).

Nutrient availability in the euphotic zone appeared to drive the variability of vertical POC fluxes in the NSCS. Based on previous results that the primary productivity and new production were determined by the availability of nutrients in the euphotic zone of the NSCS (Chen et al., 2005, 2008b, 2014), the vertical POC fluxes through a depth of 100 m should be dependent of nutrient availability, particularly the availability of N+N in the euphotic zone because of the remarkable nitrogen limitation ([N+N]/[DIP] << 16) in the NSCS (Chen et al., 2008b, 2014; Hung et al., 2020}. The nutrient supply and availability were in turn determined mainly using climatic and oceanic forcing (e.g., the winter intensification of wind-driven turbulence and vertical convection). Therefore, vertical POC fluxes were largely determined using integrated Chl-$a$ ($r = 0.8367$, $p < 0.01$) which was determined by the availability of DIN ($r = 0.9151$, $p < 0.01$) derived from the data collected in this experiment (Fig. 10). As a result, vertical POC fluxes were likely to vary with the varying hydrographic and nutrient conditions.



**Figure 10** Plots of positive correlations between integrated Chl-*a* and vertical POC fluxes (upper panel), and between DIN

inventories and Chl-*a* inventories in the euphotic zone (lower panel).





By combing the experimental and predicted data, we found that the seasonal, geographic, and ocean events affect the vertical POC fluxes (Fig. 7). Vertical POC fluxes were higher in winter than in other seasons in both the northern regime and central basin (SEATS). The flux was also slightly higher in the case influenced by an anticyclonic eddy than the one unaffected by an

eddy in winter in the northern regime. An exception to this pattern in POC fluxes occurred in summer; the POC fluxes were expected to be low, but were highly elevated due to the impact of the upwelling of internal waves. Although POC fluxes were largely predicted using empirical relationships between POC fluxes and integrated new production and Chl-*a*, the overall data indicated that the highest POC fluxes were noted in winter, followed by summer and fall. Notably, for vertical POC fluxes through a depth of 120 m collected sequentially by moored traps covering summer and winter periods, extremely low POC fluxes

were observed in summer and fall but extremely high POC fluxes were observed in winter (Fig. 11c). The exceptionally high POC fluxes in winter may be caused by the more effective trapping in catching pulsed winter blooming through the sequential and continuous collection by traps with larger trapping area (TECNICAP P.P.S. 3/3) than that through the short-term (1−3 days) collection with floating traps with smaller trapping areas in each event. The highest POC fluxes correspond to the highest POC contents (wt. %) in settling mass (Fig. 11c), indicating major biological origins of the total settling materials (%POM = %POC

× 2) in winter. The highest POC fluxes were also attributable to the prevailing northeast monsoon wind (Fig. 11a) and lowest surface temperature (Fig. 11b), which enhanced surface mixing and nutrient pumping.

**Figure 11** Temporal variability of wind speed (a), surface temperature (b), and their corresponding vertical fluxes and weight (%) of POC (c) during the period the trap was moored (from summer to winter) at a depth of 120 m on the site (18°19.661'N, 115°44.103'E) close to the SEATS station. All data were adapted from unpublished data in Tsai's thesis (Tsai, 2007).

Vertical PON and POP fluxes were relatively incomplete compared with POC fluxes that elucidated the seasonal and geographic variations because of the lack of predicted data for evaluation. However, PON and POP fluxes at a depth of 100 m



followed generally with POC-flux patterns, showing the highest values (21.2±1.68 mg N m$^{-2}$ d$^{-1}$; 1.79±0.19 mg P m$^{-2}$ d$^{-1}$) in

the summer-internal wave event and lowest values (12.1±0.47 mg N m$^{-2}$ d$^{-1}$; 0.93±0.04 mg P m$^{-2}$ d$^{-1}$) in the regular summer

season. The POC:PON ratios ranged from 5.65±0.20 at a depth of 50 m to 8.56±0.20 at a depth of 150 m, which is not quite

different from the Redfield ratio (6.6). The POC:POP ratios ranged from 152±1.57 at a depth of 50 m to 243±15.3 at a depth of

150 m, which is higher than the Redfield ratio (106) and may reflect the dominant distribution of small-size phytoplankton (Chen

et al., 2008b, 2014). The C:N and C:P ratios generally increased from a depth of 50 m to a depth of 150 m, implying the

preferential decay of POP and PON over POC.

Vertical fluxes of DOC and DON through a depth of 100 m were relatively low compared with POC and PON fluxes

because of the small vertical gradient of concentrations in surface waters. Vertical DOP fluxes were negligible because of the

insignificant concentration gradient. Despite the lack of winter data, DOC and DON fluxes were expected to increase from

summer to winter because of the summer surface accumulation caused by stratification, and the increase of downward fluxes in

winter due to the erosion of stratification.

### 4.3 Ocean-wide comparisons of active fluxes, passive fluxes, and biological pumps

Overall, the active fluxes of C, N, and P were 7.56–93.4 (mean: 37.9) mg C m$^{-2}$ d$^{-1}$, 1.06–7.26 (mean: 3.64) mg N m$^{-2}$ d$^{-1}$,

and from 0.13–0.99 (mean: 0.5) mg P m$^{-2}$ d$^{-1}$, in the NSCS (Table 4). Although most previous reports lacked data on active N

and P fluxes, our magnitudes of active fluxes of C, N, and P were considerably higher than those reported in the North Pacific

Subtropical Gyre (Hamides et al., 2009; Table 4), HOTS station (Al-Mutairi and Landry, 2001; Steinberg et al., 2008; Table 4),

Canary Island (Yebra et al., 2005; Table 4), subtropical-tropical Atlantic (Longhurst, 1990; Table 4), and Northwest Pacific

(Kobari et al., 2013; Table 4). The relatively low reported values may be attributed to two reasons, the different ocean regimes

and conditions and the other active fluxes derived only from respiratory flux. The most comparable active carbon flux was

reported by Hernández-León et al. (2019) with the total active flux (36.1±33.0 mg C m$^{-2}$ d$^{-1}$; Table 4) derived from the respiratory,

gut, excretory, and mortality fluxes in the subtropical-tropical Atlantic, These data are very close to our estimated active C fluxes,

which is likely because of the same estimation method used.

Because of the small contributions of DOC and DON fluxes to passive fluxes, our passive fluxes can be compared directly

with previous vertical fluxes of POC. The range and mean values of our data are comparable with those recorded in the same



oceanic regime (most from the SEATS station) during various periods (Chen, et al., 2008a; Ho et al., 2010; Wei et al., 2011, Cai

et al., 2015; Table 4), although the passive fluxes of N and P have not been recorded. Our data are strikingly close to the fluxes

of C, N, and P reported from the Costa-Rica-Dome upwelling system (Stukel et al., 2016; Table 4). However, our data are

apparently higher than those reported from the Northeast Pacific (Knauer et al., 1979; Table 4), BATS station (Helmke et al.,

2010; Table 4), and North Pacific Sutbtropical Gyre (Hamides et al., 2009; Table 4). This may imply that the NSCS effectively

mediates carbon transfer from the surface to the interior of the ocean.

The total export of carbon from the surface into the interior of the ocean in the South China Sea ($3.5 \times 10^6$ km$^2$) may be

extrapolated from the total BP measured in the NSCS. To a first approximation, the total export was preliminarily projected to

be 0.208 Gt C yr$^{-1}$ [(163 mg C m$^{-2}$ d$^{-1}$) × ($3.5 \times 10^6$ km$^2$) × (365 d/yr)], which is approximately 1.89% of the global annual flux

(11 Gt C yr$^{-1}$) reported by Sanders et al. (2014). This value is expected to change if more BP data are available for the SCS.

Nevertheless, the annual C flux was higher than the value reported from the North Atlantic (0.55−1.94 Gt C yr$^{-1}$; Sanders et al.,

2014) if the area of the SCS was normalized to that of the North Atlantic (43.45 km$^2$); thus, the SCS, as the largest marginal sea,

may play a more efficient role than open oceans in the transfer of atmospheric $CO_2$ into deep layers.





**Table 4** Summary and comparison of estimated active, passive (through a depth of 100 m), and total fluxes of carbon, nitrogen, and phosphorus in NSCS and other oceans

| Region | Total flux (mg m⁻² d⁻¹)[a] | | | Active flux (mg m⁻² d⁻¹) | | | Passive flux (mg m⁻² d⁻¹)[+] | | | Ref[b] |
|---|---|---|---|---|---|---|---|---|---|---|
| | C | N | P | C | N | P | C | N | P | |
| NSCS/Range | 71.9–347 | 13.0–30.5 | 1.02–2.97 | 7.56–93.4 | 1.06–7.26 | 0.13–0.19 | 65.3–255 | 11.9–23.2 | 0.89–1.98 | 1 |
| NSCS/Mean[c] | 163 | 21.2 | 1.94 | 37.9 | 3.64 | 0.5 | 125 | 17.6 | 1.44 | 1 |
| NSCS (%Total)[d] | | | | 23.30% | 17.20% | 25.80% | 76.70% | 83.00% | 74.20% | 1 |
| NSCS-basin | | | | | | | 118(summer)–209(winter) | | | 2 |
| NSCS-basin | | | | | | | 61.4(summer)–241(winter) | | | 3 |
| NSCS-basin | | | | | | | 51.6(summer)–116(winter) | | | 4 |
| NSCS-basin | | | | | | | 63.6(fall)–220(spring) | | | 5 |
| BATS | | | | | | | 29.1±14.3 (150 m) | | | 6 |
| Northeast Pacific | | | | | | | 68.4 (75m) | 5.74 (75m) | 0.43 (75 m) | 7 |
| Costa Rica Dome | | | | | | | 120±8.4 | 12.6±1.5 | 0.81±0.13 | 8 |
| N.Pacific Subtropical Gyre | 33.7 | 5.66 | 0.56 | 4.91 (14.6%)[d] | 1.46 (25.8%)[d] | 0.22 (38.3)[d] | 29.0[e] (86%)[d] | 4.2[e] (74%)[d] | 0.34[e] (61%)[d] | 9 |
| Subtropical-tropical Atlantic | | | | 2.8–8.8 (fall) | | | | | | 10 |
| | | | | 1.1–123.8 (36.1±33.0) (25-80 %)[d] | | | | | | 11 |
| HOTS (1990-1996) | | | | 3.65±2.08 | 0.63±0.36 | | | | | 12 |
| HOTS | | | | 3.65 (summer) | | | | | | 13 |
| Canary Island | | | | 8.42 (eddy) | | | | | | 14 |
| | | | | 1.85 (summer) | | | | | | |
| Northwest Pacific | | | | 2.2 | | | | | | 15 |





[a]Total flux = (active flux) + (passive flux); [b]Ref (Reference): 1 (This study); 2 (Ho et al., 2010); 3 (Wei et al., 2011); 4 (Cai et

al., 2015); 5 (Chen et al., 2008); 6 (Helmke et al., 2010); 7 (Knauer et al., 1979); 8 (Stukel et al., 2016); 9 (Hannides et al., 2009);

10 (Longhurst et al.,1990); 11 (Hernández-León et al., 2019); 12 (Al-Mutairi and Landry, 2001); 13 (Steinberg et al., 2008); 14

(Yebra et al., 2005); 15 (Kobari et al., 2013); [c]Mean: Mean value; [d]%: the percentage (fraction) of total flux; [e]29.0: the value

reported at 150 m.





### 4.4 Relative contributions of active fluxes and passive fluxes to biological pumps

Contributions of active fluxes of C, N, and P to total fluxes of C, N, and P accounted for 23.3%, 17.2%, and 25.8%, respectively (Table 4). Despite the limited data available for other oceans, in our study, the magnitude of contribution of active C flux was lower, but that of contributions of active N and P fluxes was higher than the corresponding findings by Hannides et al. (2009) in the North Pacific Subtropical Gyre (Table 4). However, the magnitude of contribution of active flux in our study was apparently lower than the range reported by Hernández-León et al. (2019; Table 4) in the subtropical-tropical Atlantic. Overall, the range of difference in total fluxes (BP) was reasonable, which may imply that our findings are reliable. The C:N and C:P ratios in the BP were 7.69 and 84.0, respectively, indicating higher C and P enrichment compared with the Redfield ratio. This may be attributed to the more pronounced enrichment in C and P in active fluxes (C:N = 10.4; C:P = 75.8) because the ratios in passive fluxes (C:N = 7.1; C:P =86.8) are close to the Redfield ratio. DVM-mediated transport may play a crucial role in the transfer of P from the surface to the mesopelagic zone.

### 5 Conclusions

To understand the strength of carbon removal from the surface to the interior of the ocean, the study of BPs is essential. Elucidating the BPs of C, N, and P in the SCS is a high research priority not only because of the limited existing data on the regimes but also for increasing the knowledges of the BP responses to changing tropical oceans. Overall, the collected and predicted data indicated that the passive fluxes of C, N, and P were seasonally variable and particularly higher in winter than in other seasons in the NSCS. The strengths of passive fluxes were estimated as 66.3–255 (mean: 125) mg C m$^{-2}$ d$^{-1}$, 11.9–23.2 (mean: 17.6) mg N m$^{-2}$ d$^{-1}$, and 0.89–1.98 (mean: 1.44) mg P m$^{-2}$ d$^{-1}$, of which the fluxes of DOC, DON, and DOP accounted for generally less than 5%. Active fluxes varied largely in coincidence with the seasonal variations of passive fluxes, ranging from 7.56 to 93.4 (mean: 37.9) mg C m$^{-2}$ d$^{-1}$, from 1.06 to 7.26 (mean: 3.64) mg N m$^{-2}$ d$^{-1}$, and from 0.13 to 0.99 (mean: 0.5) mg P m$^{-2}$ d$^{-1}$ in the NSCS. They usually account for less than one-third of the total fluxes (BPs). Both active and passive fluxes exhibited contrasting patterns between summer and winter, resulting mainly from surface warming and stratification in summer and cooling and wind-induced turbulence in pumping nutrients into the euphotic zone in winter. The increase in nutrient availability appeared to increase the primary and secondary production in tropical winter when the temperature remained sufficiently high for biological activity. In addition, the impact of anticyclonic eddy and internal-wave events on BP enhancement was pronounced in the NSCS. Overall, the active and passive fluxes were driven by nutrient availability within the euphotic layer, which was ultimately controlled by the change in internal and external forcings. To a first approximation, the SCS may effectively transfer 0.208 Gt C yr$^{-1}$ into the ocean's interior, accounting for approximately 1.89% of the global C flux.



## 6. Data availability

The data published in this contribution are largely included in this article and its supplementary materials. Additional data can

be accessed through email request to the corresponding author.

## 7. Author contribution

In this work, JJH planned and conducted the experiments and wrote the article; CHT, ZYL, SHP, LST, and YHL performed experiments including collection and analyses of hydrographic and biological pump data; YLC performed new-production

experiments and supervision.

## 8. Competing interests:

The authors declare that they have no conflict of interests

## Acknowledgements

The authors would like to thank Mrs M.-H. Huang, H.-D. Huang, and Y.-T. Yeh for their assistance in sampling and

analyses. This study was supported by the Ministry of Science and Technology, Republic of China (MOST 107-2621-M110-022, MOST 108-2611-M-110-015-, MOST 109-2611-M-110-008-) and the "Aim for the Top University Plan" of the National

Sun Yat-sen University and Ministry of Education, Taiwan, ROC (06C030203).

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
