# Peer review of "Active and passive fluxes of carbon, nitrogen, and phosphorus in the northern South China Sea"

_Biogeosciences, 2021_

## Referee Comment (RC2)

Title: Biological pumps of carbon, nitrogen, and phosphorus in the northern South China Sea
Authors: Hung and co-authors
Biogeosciences
Minor revision

General comments

   This paper presents the active and passive fluxes of carbon, nitrogen, and phosphorus in the South China Sea, and addresses the factors driving the spatial and temporal variations of biological pumps. The paper presents active fluxes which are seldom considered previously and can be useful globally on understanding the strength of carbon removal from the surface to the interior of the ocean. But before publication, some information should be clarified and some statements should be addressed.

1 The authors emphasize the importance of including active fluxes, but it is not clear why active fluxes includes gut, excretory, respiratory, and mortality fluxes by zooplankton and micronekton?

2 The authors estimate the excretory fluxes of dissolved organic C, N, and P by assuming organic products represent a constant fraction of the total amount of waste by-products released by migrators at depths. The constant fractions the authors used (0.24 for organic C, 0.53 for organic N, and 0.47 for organic P) are from references across from a long time differences (1963, 1997, 2000). It is difficult to understand the fraction of organic C is lower than those of organic N and organic P?

3 The biological pumps are higher in the study region than most of the comparison areas from the references. More statements are needed.

3 It is better to make clear when the observations were carried out, what kind of samples were collected, and water depth and trap depth of biological pumps. I try to get the related information and recognized it is so difficult.

Specific comments

Some abbreviations are not normal and used not often in the text. It is better to use the full name. For example, "Dongsha Atoll", the N:D?

P16, Line 298-300, It is not easy to understand the sentences. Please make it clear.

P18, Line 300-338, for the elemental ratios of C:N, C:P, and N:P, the summary is not consistent with the data. In fact, P is not high but low, and the ratios of C:P and N:P are not lower than the Redfield ratio.

P27, Line 490, why the respiration and gut fluxes did not include N and P fluxes?

P37, Line 614, the ratio in passive flux (C:P=86.8) is close to the Redfield ratio? Please check the number or the statement;

Some spelling and printing should be checked carefully, here are some examples:

P2, Line 40, two points "..", delete one;

P3, Line 76, add one blank in "around5000 m";

P6, Line 116, add one blank in "theassociated";

P7, Line 126, delete one blank between "methods" and "in seawater";

P8, Line 157, add one blank;

Figure 11, the observation year should be also provided. The date expression should be clear.

---

## Author Response (AR1)

**Dear Associate Editor and reviewers**                    July 7, 2021

Firstly, I would like to thank the Associate Editor and reviewers for their constructive comments on the manuscript. The revision may not be complete without reviewers' detailed comments. In order to tracking revision, all revised and/or added statements and figures have been marked with red, blue and purple colors related to comments given by reviewer #1, reviewer #2, and reviewer #3, respectively. We appreciate reviewers' (#1 & #2) valuable and science-based comments, and we are able to follow and make complete revision. We are disappointed to 3rd reviewer's decision by sticking on "uncertainty" and "eddy impacts" that are fixable to improve the quality of manuscript. It is useless to argue and make rebuttals to reviewer #3 any longer if he intends to block this first-ever data set in the South China Sea for publication. Nevertheless, it is our obligation to take Associate Editor's requests for paying attention on "uncertainty" comments given by reviewer #3. We have carefully considered the "uncertainty" of data associated mainly with temporal (seasons and extreme events) and spatial variability, rather than due to employing empirical equations in estimating respiratory and excretion fluxes that are very tiny in determining data uncertainty. Anyway, we believe that the Associate Editor will make right judgement. The followings are point by point responses to reviewers' questions.

**Replies to reviewers**

**Reviewer 1**

1. Overall, I found that the study is robust and this manuscript will be suitable for publication after some mild to moderate adjustments. For one, the use of the term "biological pumps", or "BPs" in the paper was confusing to me and I don't think it is the correct way to use that term. I think of the biological pump as a large-scale concept that is occurring in the water column; its strength can be assessed through fluxes of the C, N and P, as described in this study. Therefore, this study should use the correct terminology of what the study is really assessing (active and passive fluxes), instead of "biological pumps". I also noticed that multiple figures are distorted (i.e. stretched or compressed). Finally, the manuscript had numerous repeated grammatical errors and would thus benefit from an English proofreader.

Reply: Many thanks for comments. We agree to replace "biological pumps" as "active and passive fluxes" in the manuscript, although the "biological pump" has been widely used in published articles as an indication of total carbon flux through the euphotic zone. However, we retain the "biological pump" term in Introduction and Discussion sections for citing published articles that used the BP term in papers. In addition, we have taken care of reviewer's concerns in figures and grammatical errors.

2. Specific comments:

Line 75: A figure showing a map of the SCS would be helpful here, maybe refer to Fig. 1.

Reply: Yes, we added "see Figure 1" to the statement.

Lines 85-88: Goals of this study are confusing. Does the author mean there is limited data on the BP in the NSCS? And is the ultimate goal looking at multiple biological pumps, or the biological pump in the NCSC? Use of BP terminology is unclear.

Reply: The last paragraph was revised as followings: very few studies have addressed C, N, and P transfers from the surface to the ocean's interior. Apparently, the study of active and passive fluxes is essential and urgent to realize the states and processes of carbon fluxes in the NSCS. Thus, our ultimate goals focus primarily on understanding the current strengths of active and passive fluxes and their controlling mechanisms in the oligotrophic NSCS.

Table 1: I am confused about the sampling dates for that cruise as well, please put it in the same format as the other dates.

Reply: Table 1 was revised.

Line 129: Please provide the actual depth (i.e., pressure in water column) along with the light penetration depths here.

Reply: The depth for a specific light penetration (ex: 100%, 46%, 38%, 13%, 5% and 0.6%) is different in different sampling location. Therefore, it is inappropriate to list all depths for all sampling stations in sampling methods.

Fig. 2: Please put the season and sampling year somewhere in each subplot; I don't remember which one is which with just numbering 1-11.

Reply: Ok, we have added season/year to Figures 2.

Fig 3: In TS plot, you could label (or put box around) the different waters (subsurface, winter vs. summer waters, etc)

Reply: Ok, we have added season/year to Figures 3.

Line 300: Why is the figure not shown? Maybe just remove that text

Reply: Yes, we removed it from the text.

Figure 5 caption: "data in various expeditions"- are these the expeditions that are described here (i.e. Table 1), or are they different?

Reply: the "various" was replaced by "all".

Line 383: Term "regular summer" is unclear.

Reply: "regular summer" was replaced by "typical summer (without extreme events)".

Line 390: "an overall value" – is that the mean of both the 50 m and 100 m ratios?

Reply: Yes, we have added "averaged from 50 m and 100 m ratios." To the statement.

Table 3: Replace "predicted" with "estimated"

Reply: Yes, done.

Line 575-579: "This may imply…"; what is being said here? That the NCSC has more effective C fluxes than previous data from the open Atlantic and Pacific oceans? That was not stated and it is unclear what the sentence is saying its current form.

Reply: The statements have been revised (This may imply that the NSCS was more effective than open Atlantic and Pacific oceans in mediating POC transfer from the surface to the interior of the ocean).

Line 583: This statement about 1.89% of the global flux could be strengthened by also mentioning the area / volume of the NCSC relative to the global ocean (I am guessing it is <2%).

Reply: Yes, we have added the ocean area ratio (SCS/global ocean: 0.97%) to the text. In addition, 1.89% is replaced by 1.89±0.81%.

**Reviewer 2**

General comments:

Reply:

1. The gut, excretory, respiratory, and mortality fluxes have been well documented as major components of active flux and mentioned in the Methods (for instance, Hannides et al. 2009; Hernández-León et al., 2019). Many published articles may deal with some of them as an active flux partly because of lacking complete data, but that may result in an underestimation of active flux. To get a complete and reliable active flux, we have tried to include all components in the estimate of active flux.
2. The first reason is that organic carbon is relatively abundant compared with organic nitrogen and phosphorus; the second reason is that organic nitrogen and phosphorus are more labile than organic carbon.
3. We have tried to compare with all published data. Some of published "biological pump (BP)" papers just treat sinking fluxes (POC) as BP and such types of papers were excluded for comparison.
4. I am not so sure for reviewer's concerns. We have clearly indicated that floating traps were deployed at 50 m, 100 m, and 150 m, but the fluxes at 100 m were regarded as the sinking fluxes because of the euphotic zone being <100 m. The DOC flux was also calculated through the depth of 100 m. In terms of active

fluxes, they were estimated from the day-night difference of migrators within the top layer of 200 m (see Methods).

Specific comments

1. Ok, we used the full names (Dongsha Atoll, night/day ratio) in the text.
2. The statements have been revised as followings: The vertical distribution and acoustic scattering layers of migrators recorded at frequencies of 38 kHz (depth > 1000 m) and 120 kHz (depth approx. ~300 m), respectively, clearly had strong image layers around 400 m derived from 38 kHz data during the day and approximately 100 m derived from 120 kHz data during the night.
3. Many thanks for pinpoint error in the final statement of elemental ratios (C:N:P) in mesozooplankton. The statement has been revised.
4. Respiratory flux did not involve in N and P fluxes because respiration only release $CO_2$ (DIC) but no gas states of N and P.
5. The statement was revised as the following: the ratios are closer to the Redfield ratio in passive fluxes (C:N = 7.1; C:P =86.8) than in active fluxes.
6. Thanks for pinpointing spelling errors. We have made correction.
7. The caption of Fig. 11 was revised to show clearly the sampling period.

**Reviewer 3**

Major

- Authors here present a logic behind: what they all estimated in current work are all belonging to BP. But they don't ask readers' idea. What if some readers doesn't agree? To investigate vertical flux is important, but I don't think it is good to changing classic concepts without clarifying it.

Reply: Although the "biologic pump" was widely used to represent active and passive fluxes, we agree with the reviewer's points (and also suggests from reviewer #1) and therefore the "BP" was replaced by active and passive fluxes in the revised manuscript except for citing previous reports that used BP in their papers.

- The zooplankton-related vertical fluxes, including gut flux, respiratory flux, excretory flux, mortality flux, how about the uncertainties? I would guess the uncertainties are large. If so, some of the authors conclusion may change accordingly.

Reply: The gut flux, respiratory flux, excretory flux, and mortality flux were widely regarded as major components of active fluxes (for example, Hannides et al. 2009;

Hernández-León et al., 2019). You can find numerous papers using some of them as "active flux" regardless the incomplete estimate. Although each flux was derived from certain calculation under particular assumptions, the calculation was also based on in-situ data (for example, CTD, measured Chl-a data, and zooplankton data) rather than derived simply from modelling. The uncertainty contributed to the total vertical flux would be small because of a fraction (1/3) of active flux to the total vertical flux. The active flux in the SCS will be improved certainly if more data are available. Unfortunately, very limited data are available at the current state in the SCS.

- Another question is how to persuade readers that by equations from other sites, the zooplankton-related vertical fluxes are still valid and make sense for the SCS case? For example, the Takahashi 2009 equation is from a subarctic pacific ocean, how can it be applied in the tropical SCS? how can respiration rate be the same between sharp habitat background? Even if yes, this meant large uncertainties, how to access its impact on your presented result and conclusion? I failed to see it.

Reply: I understand reviewer's concerns. However, the used carbon respiration rate that was not a specific value reported by Takahashi 2009; rather, it was estimated from the empirical equation that was dependent on in-situ temperature and zooplankton size. Therefore, the respiration rate was determined under considering the local ocean conditions.

- It is clear zooplanktons also breath, release, eat, and excrete when they are in upper water columns. Some of them even die (e.g., be grazed by fish) when in the upper water columns. All these activities means that presence of zooplankton in the upper water column also contributes CNP to the upper water column. This is, I guess, in the reverse way the authors are discussing. I guess authors have considered and made calibration against this process. But I failed to see it. Can authors explain this more clearly? How they cancel out this part?

Reply: You don't need to worry about how zooplanktons eat, breath, and excrete in the upper water column. The logic behind the active flux was that the flux was determined by migrators (difference between night abundance and day abundance in the top layer of 200 m) that carry carbon and elements through the water column without considering the metabolic states in upper and mesopelagic zones. All zooplankton's consumption and release (i.g. POC, DOC) in the upper layer may eventually have impacts on POC sinking fluxes or DOC vertical fluxes which belong to passive fluxes.

- I have doubts in simply comparing collected zooplankton biomass between day and night net. In addition to vertical migration, how to consider zooplankton lateral migration across slope area along with tide? Quite a few sites are on slope regions as is shown in figure 1. so these sites should be under such impact.

Reply: Overly concerned. The lateral flux is ubiquitous but active flux was not determined exclusively by the state of zooplankton abundance in a specific layer. Once again, the active flux was determined by vertical migrators that were determined by a period of time during repeated trawling at night and day. In addition, the effect of lateral migration on zooplankton abundance could be same at day- and night- time if there had pronounced tidal effects.

according to the methodology, I don't think the authors present results are PON, it is acid-rinsed PN. Some ON may lost during acid rinsing process

Reply: I agree with reviewer's point. PON was replaced by PN in the manuscript.

- Vertical fluxes of DOC and DON: though I am not familiar with the way authors did the calculation, why the authors are sure the vertical flux is one-way only (from upper to bottom layer)? How if the upwelling or any other physical process that brings bottom water (hence DOC and DON) to shallower layers? I see usually DOC concentration vertically dropped from surface to bottom waters, but if there is upwell-like phenomenon, how to make sure the down-ward flux of DOM, as present in current work, exist?

Reply: Vertical fluxes of DOC and DON were determined by vertical concentration gradients and diffusivities. Any physical processes may affect DOC and DON concentration in depths but were not critical factors in determining DOM fluxes. The vertical flux was primarily determined by how DOM accumulates in the upper layers during stratification (e.g. summer) and vertical transfer during turbulent mixing in winter.

- Eddies indeed play an important role in determination of vertical fluxes. Recent work shows that in SCS, the detailed eddy information is also very important in its determination of ecosystem and biogeochemistry. At the edge of warm eddy, it can be upwelling, whereas at the edge of cold eddy, it can be downwelling [1]. The timing of eddy is also important[1]. In current work, the site and timing of eddy information is missing. So it is hard to judge the eddy contribution to vertical flux.

Reply: Many thanks for comments and providing valuable paper for reference. We have added important information of anticyclonic eddy to the Discussion section, including the section 4.1 for the description and mechanism of eddy formation and section 4.2 in elucidating the effects of eddies on vertical POC fluxes. References (Xiu et al., 2010; He et al., 2019; Zhou et al., 2020) were added for citation in the text.

- Line580 extrapolate the entire SCS via continental-slope-based data should be viewed with caution, as slope region is different from basin area in SCS.

Reply: We understand the limit in doing extrapolation. Hopefully, the data will be improved while more data become available in the SCS.

**Specific:**

Abstract: this sentence is confusing: Vertical fluxes of dissolved organic C, N, and P generally contributed to less than 5% of passive fluxes. In word, this is not logic. What do you mean?

Reply: The statement was revised (Vertical fluxes of dissolved organic C, N, and P were small ($< 5\%$) relative to passive fluxes).

Introduction

Line 49-50: This active transport may not only be important in sustaining the metabolic requirement of mesopelagic community, but also provide partial energy demand of mesopelagic ecosystem---this two aspects are the same thing.

Reply: The statement was changed (This active transport was important in sustaining the metabolic requirement of mesopelagic community through providing partial energy demand of mesopelagic ecosystem)

Lack of scientific question in the introduction part

Reply: Introduction was revised considering comments from all reviewers.

Line136 0.125 lack unit

Reply: Yes, done.

In offshore regions, SCS water is very clean and less of POM. Would it be enough to measure POC precisely based on only 1.5L of seawater? Authors should present their instrument detection limit accordingly.

Reply: This should be 1500−2500 ml after checking original data. Sorry for incorrectly applying ECS cases on the sampling statement. Thanks.

Line 227-228, Organic matter content was estimated from POC content by a factor of 2. What does this mean?

Reply: Adding (%POM = %POC $\times$ 2) to the sentence.

Line 395 Missing 'than'?

Reply: Ok, thanks.

Line 518 bracket wrong

Reply: Ok, thanks.

---

## Referee Report (RR1)

**Review of revised manuscript "Active and passive fluxes of carbon, nitrogen, and phosphorus in the northern South China Sea" (bg-2021-17)**

**General comments**:

Overall, the authors have addressed my original comments and I find the manuscript has improved. However, I have one general comment. As the paper now stands, there is little comparison of this study with the flux results from other marginal sea studies in the Discussion, even though there are different marginal seas described in the Introduction. Ultimately, the authors compare the data in this study to the Costa Rica dome, BATS and the open Pacific Ocean. Are there no other studies regarding passive and active fluxes in marginal seas that can be discussed? Particularly other Pacific marginal seas, such as the Bering Sea, Japan Sea, East China Sea and California gulf, as well as any other published data from the South China Sea. The Discussion would benefit from a more in-depth comparison of the fluxes from different marginal seas. I also noticed a few lines within the manuscript that would still require English proofreading.

**Specific comments**:

Fig 9: perhaps the datum from a different source should be labeled somehow. Also, is the black line a fit for all the points in the figure, or just data with a certain color? Please clarify in the caption.

---

## Author Response (AR2)

**Dear Associate Editor and reviewers**                                      August 11, 2021

I would like to thank the Associate Editor and reviewers again for their second comments on the revised manuscript. We have taken care of reviewers' comments and examined carefully ensuring to correct all bugs in the manuscript. The followings are second responses to reviewers' comments.

**Replies to reviewers**
**Reviewer 1**
General comments:

Overall, the authors have addressed my original comments and I find the manuscript has improved. However, I have one general comment. As the paper now stands, there is little comparison of this study with the flux results from other marginal sea studies in the Discussion, even though there are different marginal seas described in the Introduction. Ultimately, the authors compare the data in this study to the Costa Rica dome, BATS and the open Pacific Ocean. Are there no other studies regarding passive and active fluxes in marginal seas that can be discussed? Particularly other Pacific marginal seas, such as the Bering Sea, Japan Sea, East China Sea and California gulf, as well as any other published data from the South China Sea. The Discussion would benefit from a more in-depth comparison of the fluxes from different marginal seas. I also noticed a few lines within the manuscript that would still require English proofreading.

Reply: 1. We have added two reports stating active flux and total C flux from Northeast Pacific (Davison et al., 2013) and a global estimate (Hernández-León et al., 2020) for comparison. Actually, there were limited data containing both active and passive fluxes in a complete state covering various seasons and ocean regimes, particularly in the South China Sea. That may be why our data set are valuable for global C flux. 2. We have carefully examined the manuscript to ensure the quality improvement.

Specific comments:

Fig 9: perhaps the datum from a different source should be labeled somehow.

Also, is the black line a fit for all the points in the figure, or just data with a certain color? Please clarify in the caption.

Reply: We have revised the Fig. 9 caption by adding a statement (The solid black line denotes the linear regression ($r = 0.8435$, $p < 0.01$, n = 8) between INP and POC fluxes for all presented data) to complete the caption. Different icons were used to indicate different sources of data. To avoid confusion, additional labels may not be necessary for specific data.

Reviewer 3

Basically I think the authors did a good job in replying my concerns. But I still have some comments and/or suggestions before the manuscript can be considered as accepted for publication.

1 uncertainties in flux estimate.

Authors seem not able to present uncertainties due to limited data. I strongly suggest the authors clearly stated the uncertainties shortage problem in their flux estimates, to remind the readers for this point.

Reply: We have dealt uncertainty assessment. I am not sure why the reviewer claimed the shortage problem. The uncertainty (standard deviation) was derived from the spatial and seasonal (including extreme events) variability of the NSCS. The estimation (computation) of standard deviation follows exclusively the principle of statistical methods. Please see the added statements in Section 4.3 (The uncertainty of flux was mainly associated with the spatial and seasonal (including extreme events) variability in the NSCS. As active fluxes and passive fluxes may increase toward mesotrophic and eutrophic domains (Steinberg and Landry, 2017; Yebra et al., 2018; Hernández-León et al., 2019), these estimates (mean±std) may be regarded as the lower-bound fluxes under the state that the

oligotrophic regime dominates the entire region of SCS).

2 lateral migration of zooplankton or other swimmers.

In the SCS slope regions, there is already some work (Wang et al., 2019) doing the lateral migration. While their work is basically focusing on fish for their vertical migration, it also shed light in the lateral advection (as can be see from abstract). This phenomenon brings uncertainties to the active vertical flux estimates. I suggest authors cite and discuss this point with their current work.

Reply: Thanks for providing information. We have cited the reference (Wang et al., 2019) and added key points of ref. to Section 3.2.1 (The major located layers of migrators during day-time and night-time were comparable to those found for diel migrated fish in the northern slope of SCS (Wang et al., 2019) and Section 4.3 (There was an interesting report that the lateral migration of fish played an important role on determining DVM transport across the slope of NSCS (Wang et al., 2019), the impact of this issue on active fluxes is unknown in the oligotrophic ocean but this scenario warrants further study.

3 DOC and DON vertical flux

It is true that DOM vertical flux is largely constrained by water column hydrographical feature. Vertical mixing condition usually introduces surface DOM moving to deep waters. I agree that summer time is believed to be more stratified relative to winter times for the SCS, but this reply has two problems: 1) it avoids another two seasons: spring and autumn, which is hard to give a simple conclusion; 2) the meso-scale process which strongly interferes the water column vertical feature from its seasonal settings.

Again, this refer to my previous suggested literatures that discuss about the eddies process and its dynamic impact on SCS biogeochemistry. I am glad to see that the authors have cited those eddy works in their revised version.

In addition, I suggest authors consider the seasonal DOM composition difference of the SCS, as well as the spring/autumn water column vertical feature difference in response to eddies (Zhu et al., 2021).

Reply:

DOC flux contributes a very small proportion (<5%) to total vertical C flux even pronounced seasonal (eddy) variations in DOM composition. Vertical DOC flux was determined exclusively by surface accumulation and downward transport in various seasons (summer-winter) in the oligotrophic ocean. Spring and autumn are transient seasons in the SCS. Cyclonic and/or anticyclonic eddies have profound impacts on vertical POC and active fluxes by pumping nutrients into the eutrophic zone. However, eddies also lift DOC-poor water and dilute DOC concentration in upper layers and decrease likely the vertical DOC flux. We don't want to extend this arguments because of lacking data of DOM composition in our study. It is an interesting subject and the reviewer can explore for such studies.

---

## Author Response (AR3)

Dear Associate Editor:                                        August 20, 2021

   Thank you so much for pinpointing the areas throughout the manuscript for correction and improvement. We have made changes accordingly. It may not need to answer by a step-by-step manner, because we have answered all questions and examined the revised manuscript carefully again before submission. Thanks again.

Best regards,

JJ Hung